# Heterogeneous non-canonical nucleosomes predominate in yeast cells *in situ*

Zhi Yang Tan[1†], Shujun Cai[1†], Alex J Noble[2†], Jon K Chen[1†], Jian Shi[1], Lu Gan[1]*

[1]Department of Biological Sciences and Center for BioImaging Sciences, National University of Singapore, Singapore, Singapore; [2]National Resource for Automated Molecular Microscopy, Simons Electron Microscopy Center, New York Structural Biology Center, New York, United States

**Abstract** Nuclear processes depend on the organization of chromatin, whose basic units are cylinder-shaped complexes called nucleosomes. A subset of mammalian nucleosomes *in situ* (inside cells) resembles the canonical structure determined *in vitro* 25 years ago. Nucleosome structure *in situ* is otherwise poorly understood. Using cryo-electron tomography (cryo-ET) and 3D classification analysis of budding yeast cells, here we find that canonical nucleosomes account for less than 10% of total nucleosomes expected *in situ*. In a strain in which H2A-GFP is the sole source of histone H2A, class averages that resemble canonical nucleosomes both with and without GFP densities are found *ex vivo* (in nuclear lysates), but not *in situ*. These data suggest that the budding yeast intra-nuclear environment favors multiple non-canonical nucleosome conformations. Using the structural observations here and the results of previous genomics and biochemical studies, we propose a model in which the average budding yeast nucleosome's DNA is partially detached *in situ*.

*For correspondence:
lu@anaphase.org

[†]These authors contributed equally to this work

**Competing interest:** The authors declare that no competing interests exist.

## eLife assessment

This **important** paper exploits new cryo-EM tomography tools to examine the state of chromatin in situ. The experimental work is meticulously performed and **convincing**, with a vast amount of data collected. The main findings are interpreted by the authors to suggest that the majority of yeast nucleosomes lack a stable octameric conformation. Despite the possibly controversial nature of this report, it is our hope that such work will spark thought-provoking debate, and further the development of exciting new tools that can interrogate native chromatin shape and associated function in vivo.

## Introduction

Eukaryotic chromosomes are polymers of DNA-protein co–mplexes called nucleosomes. An octamer of proteins, consisting of a heterotetramer of histones H3 and H4 and two heterodimers of histones H2A and H2B, resides at the nucleosome's center (*Luger et al., 1997*). Canonical nucleosomes resemble 10-nm-wide, 6-nm-thick cylinders, and have 145–147 base pairs of DNA bent in 1.65 left-handed superhelical gyres around the histone octamer (*Zhou et al., 2019*; *Zlatanova et al., 2009*). In contrast, non-canonical nucleosomes have either partially detached DNA, partially detached histones, fewer than eight histones, or a combination of these features (*Zlatanova et al., 2009*). Both X-ray crystallography and single-particle cryo-electron microscopy (cryo-EM) have shown that reconstituted nucleosomes, either alone or within a complex, are largely canonical *in vitro* (*Zhou et al., 2019*). Nucleosome structures *in situ* inside cells remain mysterious.

**Figure 1.** Canonical nucleosomes *in vitro* and *in situ*. (**A**) Space-filling model of the reconstituted yeast nucleosome crystal structure (PDB 1ID3) (*White et al., 2001*), showing from left to right, the disc, side, and gyre views. The pseudo-dyad axis is indicated by the arrow. The DNA is rendered as light blue and the histones in the core are shaded blue (H3), green (H4), red (H2B), and yellow (H2A). (**B**) Subtomogram average of nucleosomes from wild-type (BY4741) yeast nuclear lysates. The linker DNA is indicated by the short arrows and the DNA gyre motifs are indicated by the arrowheads. (**C**) Subtomogram averages of nucleosomes in wild-type cell cryolamellae, oriented similarly to the nucleosomes in the other two panels. The upper (blue) class has more ordered linker DNA than the lower (magenta) class. Note that the subtomogram average in panel B looks different from those in panel C (and in *Figure 3* and *Figure 3—figure supplement 5*) because it is at higher resolution (18 Å vs 24 Å). The gap in the disc view of the nuclear lysate-based average is due to the lower concentration of amino acids there, which is not visible in panel A due to space-filling rendering. This gap's visibility may also depend on the contrast mechanism because it is not visible in the Volta phase plate (VPP) averages.

The online version of this article includes the following video and figure supplement(s) for figure 1:

**Figure supplement 1.** Subtomogram classification workflow.

**Figure supplement 2.** Overview of BY4741 (wild-type) nuclear lysate, defocus data.

**Figure supplement 3.** Direct 3D classification of BY4741 (wild-type) nuclear lysates.

**Figure supplement 4.** Overview of a BY4741 (wild-type) cell cryolamella, defocus data.

**Figure supplement 5.** Classification of BY4741 (wild-type) nucleosome-like particles *in situ*.

*Figure 1 continued on next page*

*Figure 1 continued*

**Figure supplement 6.** Direct 3D classification of BY4741 (wild-type) cell cryolamellae densities.

**Figure supplement 7.** Classification using the nucleosome crystal structure reference.

**Figure supplement 8.** Overview of a BY4741 (wild-type) cell cryolamella, Volta phase plate (VPP) data.

**Figure supplement 9.** Classification of nucleosome-like particles *in situ* in BY4741 (wild-type) cell Volta phase plate (VPP) data.

**Figure supplement 10.** Direct 3D classification of BY4741 (wild-type) nuclei in Volta phase plate (VPP) tomograms of cryolamellae.

**Figure supplement 11.** Overview of a BY4741 (wild-type) cell cryolamella imaged in the cytoplasm, Volta phase plate (VPP) data.

**Figure supplement 12.** Classification of BY4741 (wild-type) cell cryolamellae Volta phase plate (VPP) densities from the cytoplasm.

**Figure 1—video 1.** Direct 3D classification of BY4741 (wild-type) nucleosome-like particles in Volta phase plate (VPP) tomograms of cell cryolamellae, round 1. https://elifesciences.org/articles/87672/figures#fig1video1

**Figure 1—video 2.** Direct 3D classification of BY4741 (wild-type) nucleosome-like particles in Volta phase plate (VPP) tomograms of cell cryolamellae, round 2. https://elifesciences.org/articles/87672/figures#fig1video2

In the context of chromatin, nucleosomes are not discrete particles because sequential nucleosomes are connected by short stretches of linker DNA. Variation in linker DNA structure is a source of chromatin conformational heterogeneity (*Collepardo-Guevara and Schlick, 2014*). Recent cryo-EM studies show that nucleosomes can deviate from the canonical form *in vitro*, primarily in the structure of DNA near the entry/exit site (*Bilokapic et al., 2018*; *Fukushima et al., 2022*; *Sato et al., 2021*; *Zhou et al., 2021*). In addition to DNA structural variability, nucleosomes *in vitro* have small changes in histone conformations (*Bilokapic et al., 2018*). Larger-scale variations of DNA and histone structure are not compatible with high-resolution analysis and may have been missed in single-particle cryo-EM studies.

Molecular-resolution (2–4 nm) studies of unique objects like cells may be obtained by cryo-electron tomography (cryo-ET), a form of cryo-EM that generates 3D reconstructions called cryotomograms. These studies reveal life-like snapshots of macromolecular complexes because the samples are prepared and then imaged in an unfixed, unstained, frozen-hydrated state. Most

eukaryotic cells are too thick for cryo-ET, so thinner frozen-hydrated samples are made by cutting by cryomicrotomy or thinning by cryo-focused ion beam (cryo-FIB) milling (*Ng and Gan, 2020*; *Strunk et al., 2012*). These two approaches respectively produce cryosections and plank-like samples called cryolamellae. Subvolumes called subtomograms contain independent copies of the cells' macromolecular complexes. These subtomograms can be further studied by averaging, which increases the signal-to-noise ratio, and classification, which facilitates the analysis of heterogeneity. Large macromolecular complexes such as ribosomes and proteasomes have been identified *in situ* by 3D classification followed by comparison of the class averages to known structures – an approach called purification *in silico* (*Beck and Baumeister, 2016*).

Using the purification *in silico* approach, we previously showed that canonical nucleosomes exist in cryotomograms of yeast cell lysates *ex vivo* and in a HeLa cell cryolamella (*Cai et al., 2018a*; *Cai et al., 2018b*; *Cai et al., 2018c*). Herein, we use the term *ex vivo* to describe nucleosomes from lysates instead of the term *in vitro*, which is more commonly used to describe either reconstituted or purified mononucleosomes. However, our 3D structural analysis did not generate canonical nucleosome structures from cryosectioned fission yeast cells (*Cai et al., 2018b*). The discrepancy between nucleosome class averages *ex vivo* and *in situ* could have either technical or biological origins. As a biological explanation for the absence of canonical nucleosome class averages *in situ*, we hypothesized that yeast nucleosomes are either conformationally or constitutionally heterogeneous.

In this work, we test this heterogenous-nucleosome hypothesis by using cryo-ET to image both wild-type cells and strains that have nucleosomes bearing GFP as a density tag. We use the budding yeast *Saccharomyces cerevisiae*, herein called yeast, because it has only two copies of each histone gene and because it is more amenable to gene editing. Our work compares the chromatin in both lysates and thin cellular cryo-EM samples prepared primarily by cryo-FIB milling. To obtain more information about nucleosomes *in situ*, we create one strain in which H2A-GFP is the sole source of H2A, meaning that the nucleosomes are expected to project one or two extra densities from their surface. Canonical nucleosomes are abundant in nuclear lysates and, in the H2A-GFP-expressing strain's nuclear lysates, have extra densities consistent with GFP. In contrast, canonical nucleosomes account for less than 10% of the expected number of total nucleosomes in wild-type cell cryolamellae. Furthermore, neither canonical nucleosomes nor nucleosome-like particles with extra protruding densities were detected in the H2A-GFP-expressing strain. These findings suggest that the yeast intracellular environment disfavors the canonical nucleosome conformation.

## Results

### Canonical nucleosomes are abundant in wild-type yeast lysates

The crystal structure of the reconstituted yeast nucleosome (*White et al., 2001*) shows a canonical structure that is largely indistinguishable from the first published one (*Luger et al., 1997*). To describe the various views of the nucleosome, herein we use the compact nomenclature introduced by *Zhou et al., 2019*: the disc view is along the superhelical axis, and the gyre view is along the pseudo-dyad axis (*Figure 1A*). The side view, which was not defined by Zhou *et al.*, is orthogonal to both the disc and gyre views.

The original RELION subtomogram analysis workflow (*Bharat and Scheres, 2016*) involves 2D classification, followed by 3D classification (*Figure 1—figure supplement 1*, panel A), whereas in the alternative approach, 2D classification is bypassed and subtomograms are subjected directly to 3D classification (*Figure 1—figure supplement 1*, panel B); for brevity, we term this alternative method 'direct 3D classification'. Direct 3D classification is limited by computer hardware (*Kimanius et al., 2016*), but can detect more canonical nucleosomes (*Cai et al., 2018a*); see the Materials and methods for more details. In this study, the original workflow is used on a subset of samples to show example 2D class averages for comparison with other studies. However, the conclusions in this paper are drawn from direct 3D classification.

Our previous subtomogram analysis of nuclear lysates (*Cai et al., 2018c*) revealed that nucleosomes from the wild-type strain YEF473A (*Bi and Pringle, 1996*) adopt the canonical structure *ex vivo*. We repeated this experiment on the strain BY4741 (*Brachmann et al., 1998*), which serves as the wild-type and parent strain for the histone-GFP tagging mutants described later. In this experiment, yeast nuclei are isolated, lysed, then deposited on an EM grid. Cryotomograms of BY4741 nuclear lysates

reveal the crowded nucleosome-like particles and other cellular debris (*Figure 1—figure supplement 2*). Next, we template matched for nucleosome-like particles using a nucleosome-sized featureless cylinder as a reference. Direct 3D classification (*Figure 1—figure supplement 3*, panels A and B) followed by 3D refinement produced an 18 Å resolution subtomogram average of a BY4741 canonical nucleosome class (*Figure 1—figure supplement 3*, panel C). The subtomogram average of wild-type yeast nucleosomes in nuclear lysates has longer linker DNA (*Figure 1B*) than the crystal structure, which was reconstituted using a 146 bp DNA fragment. Because the nucleosome-repeat length of budding yeast chromatin is ~168 bp (*Brogaard et al., 2012*), this extra length of DNA may come from an ordered portion of the ~22 bp linker between adjacent nucleosomes.

## Canonical nucleosomes are rare in wild-type cells *in situ*

Cryo-FIB milling is a compression-free method to thin frozen-hydrated cells (*Hayles et al., 2007*; *Mahamid et al., 2015*; *Marko et al., 2006*; *Medeiros et al., 2018*; *Rigort et al., 2010*; *Villa et al., 2013*). This technique uses a beam of gallium ions to thin a cell under cryogenic conditions, producing a frozen-hydrated plank-like cellular sample called a cryolamella. Canonical nucleosomes are detectable in a HeLa cell cryolamella (*Cai et al., 2018a*), meaning that cryo-FIB milling does not grossly perturb canonical nucleosomes *in situ*. We prepared cryolamellae of wild-type BY4741 cells and then collected defocus phase-contrast tilt series. Cryotomograms of yeast cryolamellae showed that nuclei were packed with nucleosome-like particles (*Figure 1—figure supplement 4*). Two-dimensional class averages of template-matched nucleosome-like particles reveal densities that have the approximate size and shape of nucleosomes (*Figure 1—figure supplement 5*, panel A). We then subjected the particles belonging to the most nucleosome-like 2D class averages to 3D classification, following the original RELION classification workflow. However, none of these 3D class averages resemble canonical nucleosomes (*Figure 1—figure supplement 5*, panel B). While many of the nucleosome-like class averages have dimensions similar to the canonical nucleosome, none of them have densities that resemble the distinctive 1.65 left-handed gyres of DNA. To rule out the possibility that canonical nucleosomes were missed during 2D classification, we performed direct 3D classification using 100 classes. None of the resultant 100 class averages resemble a canonical nucleosome (*Figure 1—figure supplement 6*).

To determine if our template-matching and classification workflow missed the canonical nucleosomes, we re-did both template matching and 3D classification of BY4741 wild-type yeast cryolamellae with an intentionally biased reference. Instead of a featureless cylinder, we used the yeast nucleosome crystal structure (*White et al., 2001*) as the reference (*Figure 1—figure supplement 7*, panel A). If canonical nucleosomes are abundant in yeast, they should be detected as a class average that resembles a low-resolution nucleosome crystal structure. No canonical nucleosome class averages were seen in this control experiment (*Figure 1—figure supplement 7*, panels B and C).

Our previous study of a HeLa cell (*Cai et al., 2018a*), which detected canonical nucleosomes, used a Volta phase plate (VPP). VPP data has more low-resolution contrast than defocus phase-contrast data. To test if canonical nucleosomes in yeast cryolamellae are detectable in VPP data, we recorded VPP tilt series and reconstructed tomograms of BY4741 cell cryolamellae (*Figure 1—figure supplement 8*). Subtomogram analysis of the VPP tomograms by 2D classification (*Figure 1—figure supplement 9*, panel A), followed by 3D classification did not reveal a canonical nucleosome class average in BY4741 (*Figure 1—figure supplement 9*, panel B). When we performed direct 3D classification using 100 classes, we detected one class average that resembles a canonical nucleosome (*Figure 1—figure supplement 10*, panel A, *Figure 1—video 1*). A second round of classification revealed two types of class averages, one of which resembles a slightly elongated nucleosome (*Figure 1—figure supplement 10*, panel B, *Figure 1—video 2*). Refinement of these two types of nucleosomes produced 24 Å resolution averages that differ in the amount of linker DNA visible (*Figure 1C* and *Figure 1—figure supplement 10*, panel C). The class average that has more ordered linker DNA vaguely resembles the chromatosome, a form of the nucleosome that has linker DNA crossing at the entry-exit site and in contact with a linker histone (*Bednar et al., 2017*; *Zhou et al., 2015*). However, the DNA does not appear to cross at the DNA entry-exit site, and, at the present resolution, it is not possible to determine if the linker histone is present. Unlike plunge-frozen complexes, which interact with the air-water interface and have biased orientations (*Noble et al., 2018*), complexes *in situ* are not subject to such biases. Visualization of the angular distribution of the canonical nucleosomes shows that the disc views

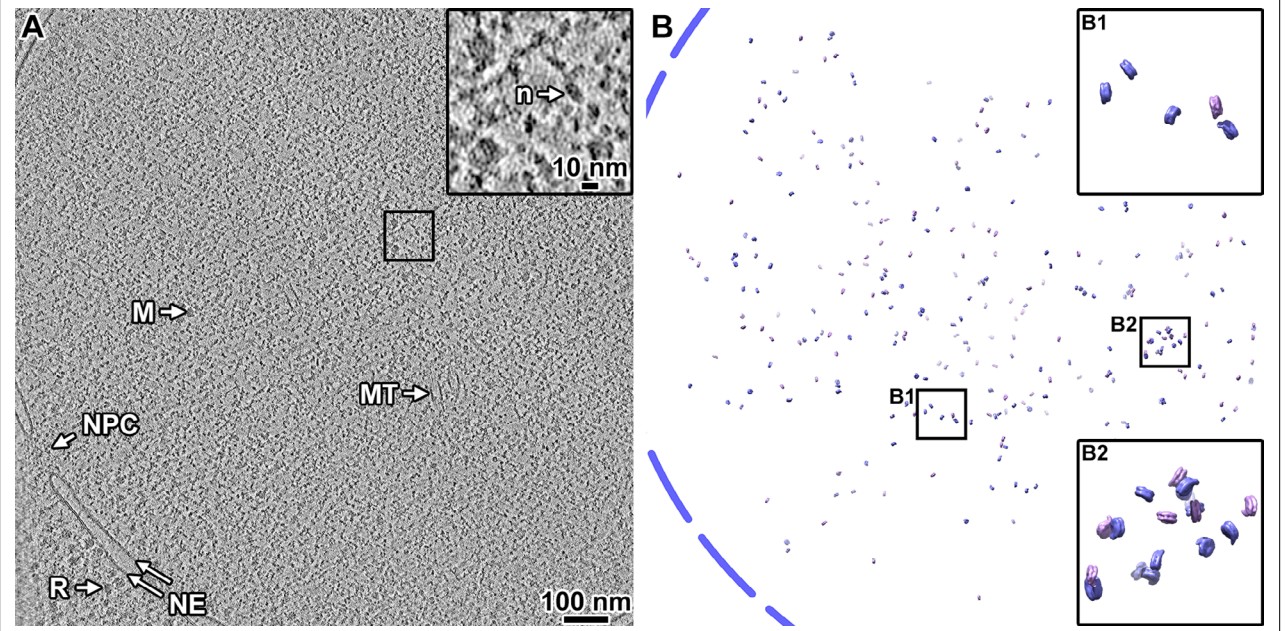

**Figure 2.** Canonical nucleosomes are a minority of the expected total in wild-type cells. (**A**) Volta phase plate tomographic slice (12 nm) of a BY4741 cell cryolamella. Large subcellular structures are labeled: nuclear pore complex (NPC), nuclear megacomplex (M), nuclear microtubule (MT), nuclear envelope (NE), and ribosome (R). The inset is a fourfold enlargement of the boxed area, and a nucleosome-like particle (n) is indicated. (**B**) Remapped model of the two canonical nucleosome class averages in the tomogram from panel A: the class averages were oriented and positioned in the locations of their contributing subtomograms. The approximate location of the nuclear envelope is indicated by the blue dashed line. The insets B1 and B2 show fourfold enlargements of the corresponding boxed areas. Note that the remapped model projects the full 150 nm thickness of this cryolamella. In this tomogram, we estimate there are ~7600 nucleosomes (see Materials and methods on how the calculation is done), of which 297 are canonical structures. Accounting for the missing disc views, we estimate there are ~594 canonical nucleosomes in this cryolamella (<8% the expected number of nucleosomes).

are undersampled and likely missed by the classification analysis (*Figure 1—figure supplement 10*, panel D), which we also observed in our analysis of HeLa nucleosomes *in situ* (*Cai et al., 2018a*). This missing hemisphere of views results in roughly half of the canonical nucleosomes going undetected. In summary, the use of VPP and relatively thin (≤160 nm) cryolamellae made it possible to detect two canonical nucleosome classes that differ in the amount of ordered linker DNA, similar to what we observed in a HeLa cell (*Cai et al., 2018a*).

Following our previous work (*Cai et al., 2018a*), we performed a negative control by analyzing a tomogram of a region in the cytoplasm (*Figure 1—figure supplement 11*), which does not have any nucleosomes. We performed template matching using the same reference as for our analysis of nuclei and then direct 3D classification into 100 classes. None of the resultant class averages resemble a canonical nucleosome (*Figure 1—figure supplement 12*), confirming that our analysis was not biased by the cylindrical reference.

Our classification detected only 769 canonical nucleosomes. If we account for the undersampling of disc views (*Figure 1—figure supplement 10*, panel D), we estimate there are ~1500 canonical nucleosomes detected in the five tomograms. In comparison, we estimate that in the single HeLa cell cryolamella cryotomogram (*Cai et al., 2018a*), there were more than 2000 nucleosomes in a nucleus volume ~1/6th of the total analyzed here in BY4741. The percentage of HeLa nucleosomes that are non-canonical *in situ* is unknown and will require further study. To visualize the distribution of canonical nucleosomes, we remapped the two class averages back into their positions in the original tomogram that contains the largest number of canonical nucleosomes (*Figure 2*). The remapped model shows that the canonical nucleosomes are scattered throughout the sampled nuclear volume. There are no large clusters of canonical nucleosomes like what we saw near the nuclear envelope of a HeLa cell. Using experimentally determined values for nucleosome number and chromatin volume (*Oberbeckmann et al., 2019*; *Uchida et al., 2011*), the tomograms we analyzed are expected to hold 25,000 nucleosomes. Therefore, the nucleosomes (both canonical and non-canonical) should pack

with an order-of-magnitude higher density than visualized in our remapped model (*Figure 2B*). In summary, our calculations suggest that the vast majority (>90%) of the nucleosomes in BY4741 yeast are non-canonical.

## Histone GFP tagging and visualization *ex vivo*

Our previous cryo-ET analysis of nucleosomes in a HeLa cell revealed that subtomogram 3D classification is sensitive to features much smaller than the nucleosomes, as evidenced by the separation of canonical nucleosome class averages that differ by ~10 bp of linker DNA near the dyad (*Cai et al., 2018a*). Furthermore, studies of flagella (*Oda and Kikkawa, 2013*) and pilus machines (*Chang et al., 2016*) showed that subtomogram averages of complexes *in situ* can reveal either the presence or absence of protein densities as small as fluorescent proteins. These observations led us to attempt to use a GFP tag to facilitate nucleosome identification *in situ*. Our strategy is to compare subtomogram averages of nucleosome-like particles in strains that express only wild-type histones versus those that express GFP-tagged histones. Note that this tagging strategy did *not* work as intended because we could not detect tagged nucleosome 3D classes *in situ*. However, this negative result provided an important clue about the nature of nucleosomes inside yeast cells (see below).

Histones can accept a genetically encoded GFP tag at either the N- or C-terminus. An N-terminal GFP tag is not expected to be visible in subtomogram averages because it would be separated from the histone's globular domain by the long, flexible N-terminal 'tail'. Therefore, we fused GFP to the histone C-terminus, which does not have a flexible tail (*Figure 3—figure supplement 1*, panels A and B) and we further confined the GFP by eliminating the peptide linker that is included in popular GFP-tagging modules (*Figure 3—figure supplement 1*, panel C). *S. cerevisiae* has two copies of each histone gene (*Figure 3—figure supplement 1*, panel D), which are arranged as gene pairs. The H2A and H2B genes are arranged as gene pairs *HTA1-HTB1* and *HTA2-HTB2* (*Hereford et al., 1979*). To maximize our chances of detecting nucleosome class averages that have an extra density, we first sought to create strains in which a histone-GFP fusion is the sole source of one class of histones (*Figure 3—figure supplement 1*, panels E and F). We deleted the entire *HTA2-HTB2* gene pair to prevent its amplification as circular DNA molecules by the flanking retrotransposon elements (*Libuda and Winston, 2006*); the resulting strain is called LGY0012. Next, we inserted the GFP gene at the 3' end of *HTA1*, without a linker, to generate LGY0016, making H2A-GFP the sole source of H2A. We confirmed LGY0016's genotype by PCR analysis (*Figure 3—figure supplement 2*, panels A and B), Sanger sequencing, and immunoblots using anti-H2A or anti-GFP antibodies (*Figure 3—figure supplement 2*, panel C). Accordingly, the LGY0016 nuclei showed bright fluorescence (*Figure 3—figure supplement 2*, panel D). We also constructed the strain LGY0015, which expresses both a H2B-GFP fusion without a linker peptide (*Figure 3—figure supplement 2*, panels E–H) and an untagged copy of H2B. We were unable to create strains that have either H2B-GFP, H3-GFP, or H4-GFP as the sole H2B, H3, and H4 sources, respectively (see H3- and H4-tagging experiments below). Consistent with this low tolerance for a GFP-tagged histone as the sole source of a histone type, the LGY0016 doubling time is ~50% longer than for wild-type BY4741 (130 min versus 85 min) in rich media.

We next performed cryo-ET of the nuclear lysates of LGY0016 and LGY0015 cells (*Figure 3—figure supplements 3 and 4*). The 2D class averages of LGY0016 and LGY0015 nuclear lysates resemble those seen in single-particle cryo-EM studies of reconstituted nucleosomes (*Chua et al., 2016*), though with lower-resolution features (*Figure 3A*, *Figure 3—figure supplement 5*, panel A). Note that 2D classification uses a circular mask, meaning that it will not bias the shape of the resultant class averages to resemble, for example, the double-lined motifs seen in nucleosome side and gyre views. To increase the number of detected nucleosomes, we used direct 3D classification into 40 classes. We obtained canonical nucleosome 3D class averages this way in the lysates of both strains (*Figure 3—figure supplements 6 and 7*; *Figure 1—videos 1 and 2*). These class averages have the unmistakable structural motifs of canonical nucleosomes, such as a 10-nm-diameter, 6-nm-thick cylindrical shape, and the left-handed path of the DNA densities. There is more DNA than the crystal structure's 1.65 gyres because lysate chromatin samples have linker DNA. All these properties are consistent with the subtomogram analysis of nucleosomes from nuclear lysates of wild-type strains BY4741 (*Figure 1B*) and YEF473A (*Cai et al., 2018c*).

Subsequent classification rounds revealed nucleosome class averages that have an extra density projecting from one or both faces (*Figure 3—figure supplement 6*, panel B, and *Figure 3—figure*

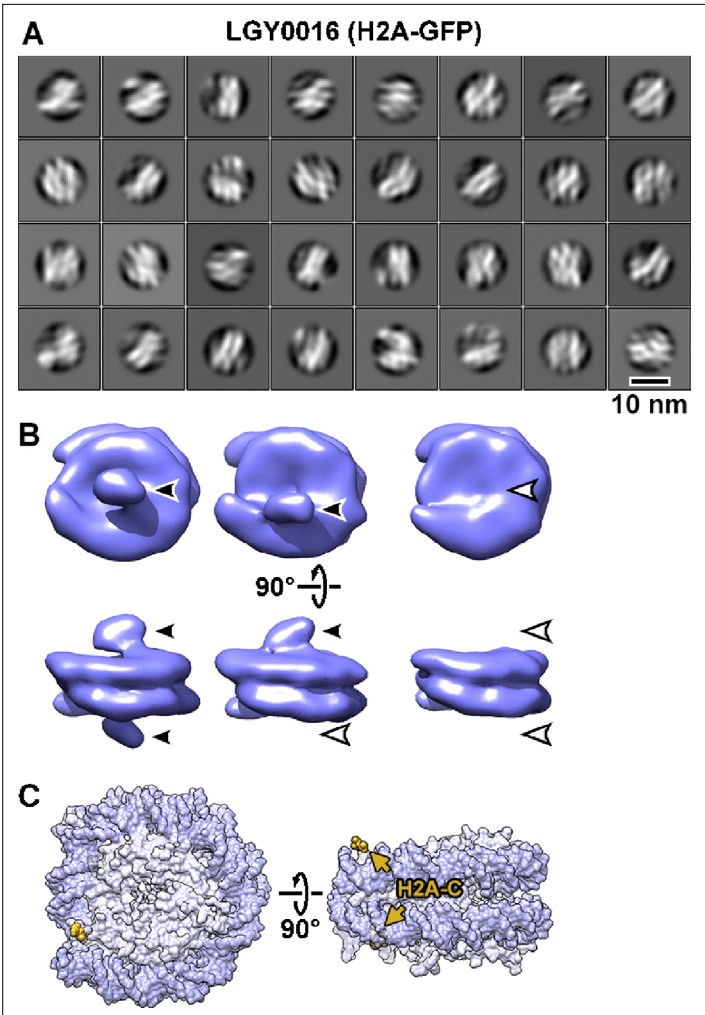

**Figure 3.** Visualization of GFP-tagged nucleosomes *in vitro*. (**A**) Example 2D class averages of nucleosome-like particles that were template-matched with a featureless cylinder reference. Out of 88,896 template-matched particles, 66,328 were retained after 2D classification. (**B**) Class averages (3D) of nucleosomes from nuclear lysates. Solid arrowheads indicate the extra (GFP) densities. Open arrowheads indicate the positions that lack this density. These class averages were obtained after classification directly from subtomogram averaging, without an intervening 2D classification step. (**C**) The approximate positions of the H2A C-termini are rendered in yellow and indicated by arrows in the crystal structure of the yeast nucleosome (*White et al., 2001*). Note that in the crystal structure, fewer of the H2A C-terminal amino acids were modeled than for H2B, meaning that the H2A C-terminus is not perfectly ordered. To facilitate comparison, this structure is oriented like the class averages in panel B. The nucleosome densities in panel B are longer along the pseudo-dyad axis (horizontal) because they have linker DNA, which is absent in the nucleosome crystal structure.

The online version of this article includes the following video, source data, and figure supplement(s) for figure 3:

**Figure supplement 1.** Strategy to tag H2A with GFP.

**Figure supplement 2.** Experimental verification of H2A-GFP and H2B-GFP tagging.

**Figure supplement 2—source data 1.** Agarose gel of PCR amplicons expected (or not) from LGY0016 genomic DNA, in which the HTB2-HTA2 locus is deleted and the HTA1 locus is tagged with GFP.

**Figure supplement 2—source data 2.** Immunoblot analysis of strains LGY0016 and LGY0012 with α-GFP antibody.

**Figure supplement 2—source data 3.** Immunoblot analysis of strains LGY0016 and LGY0012 with α-H2A antibody.

**Figure supplement 2—source data 4.** Immunoblot analysis loading control of strains LGY0016 and LGY0012 with α-H3 antibody.

*Figure 3 continued on next page*

*Figure 3 continued*

**Figure supplement 2—source data 5.** DIC and GFP fluorescence confocal microscopy for LGY0016 cells.

**Figure supplement 2—source data 6.** Agarose gel of PCR amplicons expected from LGY0015 genomic DNA, in which the HTB1 locus is tagged with GFP.

**Figure supplement 2—source data 7.** Immunoblot analysis of strain LGY0015 with α-GFP antibody.

**Figure supplement 2—source data 8.** Immunoblot analysis of strain LGY0015 with α-H2B antibody.

**Figure supplement 2—source data 9.** Immunoblot analysis loading control of strain LGY0015 with α-H3 antibody.

**Figure supplement 2—source data 10.** DIC and GFP fluorescence confocal microscopy for LGY0015 cells.

**Figure supplement 3.** Overview of LGY0016 (H2A-GFP) nuclear lysate, defocus data.

**Figure supplement 4.** Overview of LGY0015 (H2B, H2B-GFP) nuclear lysate, defocus data.

**Figure supplement 5.** Classification of LGY0015 (H2B, H2B-GFP) nuclear lysates.

**Figure supplement 6.** Direct 3D classification of LGY0016 (H2A-GFP) nuclear lysates.

**Figure supplement 7.** Direct 3D classification of LGY0015 (H2B, H2B-GFP) nuclear lysates.

**Figure 3—video 1.** Direct 3D classification of LGY0016 (H2A-GFP) lysate nucleosomes, round 1. https://elifesciences.org/articles/87672/figures#fig3video1

**Figure 3—video 2.** Direct 3D classification of LGY0016 (H2A-GFP) lysate nucleosomes, round 2. https://elifesciences.org/articles/87672/figures#fig3video2

---

*supplement 7*, panel B). One example of each class (zero, one, or two extra densities) was refined to ~25 Å resolution (*Figure 3B*, *Figure 3—figure supplement 5*, panel B). The position of the extra density is consistent with the H2A C-terminus being closer to the DNA entry-exit point (*Figure 3C*) and the H2B C-terminus being far from the DNA entry-exit point (*Figure 3—figure supplement 5*, panel C). In principle, LGY0015 cells can assemble nucleosomes that have two copies of H2B-GFP. The absence of LGY0015 nucleosome classes with two densities suggest that they are either unstable or too rare to detect by 3D classification. We focused our *in situ* cryo-ET analysis on LGY0016 cells because the GFP tags are easier to recognize on nucleosomes from nuclear lysates of this strain.

## Canonical nucleosome classes are not detected in LGY0016 cells *in situ*

In an attempt to detect more yeast nucleosomes *in situ*, we did three types of imaging experiments on LGY0016 cells. We performed cryo-ET of cell cryolamellae with and without the VPP (*Figure 4—figure supplements 1 and 2*) and we also imaged cryosections with the VPP (*Figure 4—figure supplement 3*). The cryolamellae benefit from the absence of compression artifacts while the cryosections benefit from being thinner on average. We performed template matching and then subjected the hits directly to 3D classification, which was needed to detect canonical nucleosomes in BY4741 above. The 3D class averages from all three samples resembled neither canonical nucleosomes nor cylindrical bodies with one or more protruding densities that is expected from the lysate samples (*Figure 4*, *Figure 4—figure supplements 4 and 5*, *Figure 4—video 1*). Some class averages have two linear motifs that resemble the double DNA gyres opposite the DNA entry-exit point, but these DNA-like densities do not go 1.65 times around the center of mass as expected of canonical nucleosomes (*Luger et al., 1997*). Small variations in the classification parameters such as mask size, class number did not reveal any canonical nucleosome classes in LGY0016 cell cryotomograms. We therefore conclude that canonical nucleosomes are rare inside budding yeast nuclei and that non-canonical nucleosomes are the vast majority. At present, we do not know what the non-canonical nucleosome structures are, meaning that we cannot even determine if one non-canonical structure is the majority. Until we know the non-canonical nucleosomes' structures, we will use the term non-canonical to describe all the nucleosomes that do not have the canonical (crystal) structure.

## Ribosome control for sample, data, and analysis pathologies

We may have missed canonical nucleosomes if there were pathologies with either our cryolamellae, data, or workflow, which would result in grossly misclassified and misaligned particles. To test this hypothesis, we performed template matching, 3D classification, and alignment on cytoplasmic ribosomes. Because ribosomes are so large (>3 MDa) and have been studied extensively *in situ*, any of the pathologies listed above would result in either the absence of ribosome class averages or

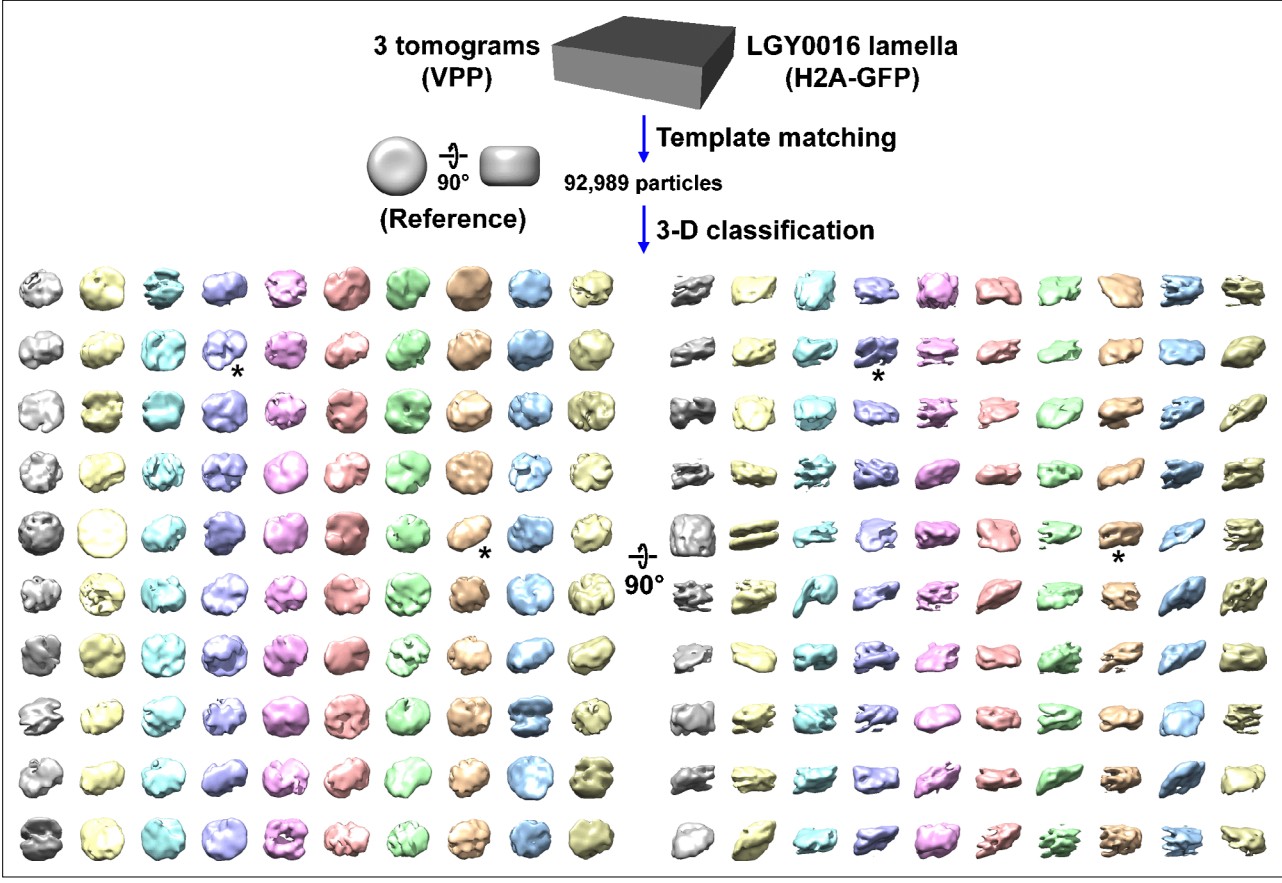

**Figure 4.** Canonical nucleosome class averages are not detected in LGY0016 (H2A-GFP) cells *in situ*. Class averages (3D) of nucleosome-like particles in Volta phase plate (VPP) cryotomograms of LGY0016 cryolamellae. The starred classes have two linear motifs. *Figure 4—video 1* shows the progress of this classification job.

The online version of this article includes the following video, source data, and figure supplement(s) for figure 4:

**Figure supplement 1.** Overview of a LGY0016 (H2A-GFP) cell cryolamella, defocus data.

**Figure supplement 2.** Overview of a LGY0016 (H2A-GFP) cell cryolamella, Volta phase plate (VPP) data.

**Figure supplement 3.** Overview of a LGY0016 (H2A-GFP) cell cryosection, Volta phase plate (VPP) data.

**Figure supplement 4.** Direct 3D classification of LGY0016 (H2A-GFP) cell cryolamellae densities.

**Figure supplement 5.** Direct 3D classification of LGY0016 (H2A-GFP) cell cryosection densities in Volta phase plate (VPP) data.

**Figure supplement 6.** Control Volta phase plate (VPP) subtomogram analysis of ribosomes *in situ*.

**Figure supplement 7.** Verification of (H3, H3-GFP) strains.

**Figure supplement 7—source data 1.** Agarose gel of PCR amplicons expected from LGY0002, LGY0007, and LGY0070 genomic DNA, in which the HHT1 locus is tagged with GFP, and LGY0071 genomic DNA, in which the HHF1 locus is tagged with GFP.

**Figure supplement 7—source data 2.** Immunoblot analysis of strains LGY0002, LGY0007, LGY0070, and LGY0071 with α-GFP antibody.

**Figure supplement 7—source data 3.** Immunoblot analysis of strains LGY0002, LGY0007, and LGY0070 with α-H3 antibody.

**Figure supplement 7—source data 4.** Immunoblot analysis loading control of strains LGY0002, LGY0007, and LGY0070 with α-H4 antibody.

**Figure supplement 7—source data 5.** DIC and GFP fluorescence confocal microscopy for LGY0002 cells.

**Figure supplement 7—source data 6.** DIC and GFP fluorescence confocal microscopy for LGY0007 cells.

**Figure supplement 7—source data 7.** DIC and GFP fluorescence confocal microscopy for LGY0070 cells.

**Figure supplement 8.** Verification of (H4, H4-GFP) strain.

**Figure supplement 8—source data 1.** Immunoblot analysis of strain LGY0071 with α-H4 antibody.

**Figure supplement 8—source data 2.** Immunoblot analysis loading control of strain LGY0071 with α-H3 antibody.

**Figure supplement 8—source data 3.** DIC and GFP fluorescence confocal microscopy for LGY0071 cells.

*Figure 4 continued on next page*

extremely low resolution. Using our control tomogram of the cytoplasm, which is densely packed with ribosomes, we obtained a subtomogram average at ~33 Å resolution based on the Fourier shell correlation (FSC)=0.5 cutoff criterion (or 28 Å using the FSC = 0.143 criterion), from 1150 particles (*Figure 4—figure supplement 6*). As a comparison, we simulated densities with the yeast ribosome crystal structure between 15 Å and 30 Å resolution and found that our average has density features consistent with this resolution range. The resolutions of our nucleosome and ribosome averages (~24 Å and 33 Å) are comparable to recent *in situ* subtomogram averages using similar numbers of particles (~500–1500) (*Laughlin et al., 2022*; *van den Hoek et al., 2022*). Therefore, our cryolamellae, data, and workflow do not show evidence of pathologies.

## GFP tagging of H3 and H4 also does not reveal nucleosomes

In cryolamellae of LGY0016 cells, the absence of nucleosome-like class averages that have an extra density bump suggested that previously unappreciated properties of the H2A-H2B heterodimer make this histone pair a poor candidate for the GFP fusion tag strategy (see Discussion). We therefore revisited GFP tagging of H3 and H4, which are found in all known and speculated forms of nucleosomes and non-canonical nucleosomes (*Zlatanova et al., 2009*). Because all attempts to make H3- or H4-GFP 'sole source' strains failed, we tested strains that had one wild-type copy and one GFP-tagged copy of one of these histones. We tagged members of the *HHF1-HHT1* gene pair, which encode histone H4 (*HHF1*) and H3 (*HHT1*). *HHF1-HHT1* is flanked by the same transposon elements as *HTA2-HTB2*, so gene amplification here would not result in the wild-type copy outnumbering the tagged copy. The genotypes and phenotypes were verified by confirmation PCRs, western blots, Sanger sequencing (not shown), and fluorescence confocal microscopy (*Figure 4—figure supplements 7 and 8*). We created H3-GFP strains without a linker (LGY0007) and with linker sequences RIPGLIN (LGY0002) and GGSGGS (LGY0070); this latter linker was introduced previously (*Verzijlbergen et al., 2010*). For H4, we could only obtain a strain with a H4-GFP-expressing strain by using the flexible GGSGGS linker (LGY0071; *Figure 4—figure supplement 8*). Flexible linkers are not expected to facilitate tag-based identification because the tag can occupy a much larger volume and get blurred out as a result but was the only one tolerated in our H4 tagging attempts.

We next prepared nuclear lysates of each of these strains, performed cryo-ET, and subjected them to direct 3D classification analysis (*Figure 4—figure supplements 9–12*). Like the other strains, the lysates of each of these four strains had large numbers of canonical nucleosomes (*Figure 4—figure supplements 9–12*, panel A of each). Of the canonical nucleosomes, we were only able to detect class averages that have the extra density in the LGY0007 nuclear lysates (*Figure 4—figure supplement 9*); the nuclear lysates of the other strains do not have nucleosome class averages with a GFP density. Finally, to test if nucleosomes, canonical or non-canonical, with the GFP density bump, could be detected *in situ*, we performed cryo-ET of LGY0007 cell cryolamellae with the VPP (*Figure 4—figure supplement 13*). We then performed template matching and subjected the hits to direct 3D classification analysis. Compared to the BY4741 (wild-type) cell cryolamella VPP class averages (*Figure 1—figure supplement 10*, panel A), the 3D class averages in LGY0007 cell cryolamellae resemble neither canonical nucleosomes nor cylindrical bodies with a protruding density (*Figure 4—figure supplement 14*). Therefore, H3- and H4-GFP fusions cannot be used to detect non-canonical nucleosomes and much more work needs to be done to identify non-canonical nucleosomes.

## Discussion

Cryo-EM of cells has made the structural analysis of chromatin organization *in situ* feasible. The earliest *in situ* studies were done with projection cryo-EM images and 2D Fourier analysis of cryo-sectioned cells, which revealed that long-range order is absent in chromatin *in situ* (*Eltsov et al., 2008*; *McDowall et al., 1986*). Cryo-ET studies later showed that short-range order is present in isolated chicken erythrocyte nuclei (*Scheffer et al., 2011*), but absent in picoplankton and budding yeast (*Chen et al., 2016*; *Gan et al., 2013*). These early cryo-ET studies revealed little about nucleosome structure *in situ* because, as Eltsov *et al.* observed, the nucleosomes in those samples appeared like smooth ellipsoids (*Eltsov et al., 2018*). Recent cryo-ET studies using state-of-the-art electron-counting cameras and energy filters have revealed that in cryotomographic slices of cell nuclei, there are densities that resemble nucleosome side and gyre views in which gyre-like features are resolved (*Cai et al., 2018a*; *Cai et al., 2018b*; *Eltsov et al., 2018*). This higher-fidelity data made it possible to use 3D classification to detect canonical nucleosomes in nuclear envelope-associated chromatin in a HeLa cell (*Cai et al., 2018a*).

Our use of 3D classification detected canonical nucleosomes in wild-type yeast nuclear lysates and canonical nucleosomes both with and without GFP tags in nuclear lysates of some strains that bear histone-GFP fusions. We also detected a canonical nucleosome class average in wild-type cell cryolamellae imaged with a VPP. The *in situ* detections suggest that there are only ~1500 canonical nucleosomes (taking account of the undersampling of nucleosomes in the disc view) out of the 25,000 nucleosomes expected of the total sampled nuclear volume. Note that the percentage of canonical nucleosomes in lysates cannot be accurately estimated because we cannot determine how many nucleosomes in total are in each field of view. The estimates from the cryolamellae are more reliable because the expected numbers of nucleosomes (canonical or not) can be estimated from genomics, biochemical, and X-ray tomography data (see Materials and methods). When we analyzed LGY0016 cell cryolamellae, in which H2A-GFP is the sole source of H2A, we did not detect canonical nucleosome classes or any cylindrical nucleosome-sized structures that have extra density bumps. Likewise, analysis of cryolamellae of LGY0007 did not reveal either a canonical nucleosome-like class average or any cylindrical nucleosome-sized density with an extra density bump, even though H3-GFP should be incorporated into the H3-H4 tetramer, the central component of the nucleosome.

The absence of either a canonical nucleosome-like class average that has an extra density bump, or a cylinder with an extra density bump in the LGY0016 strain suggests that the H2A-H2B heterodimer is mobile *in situ*. By 'mobility', we are not implying that H2A-H2B is dissociated. We mean that H2A-H2B

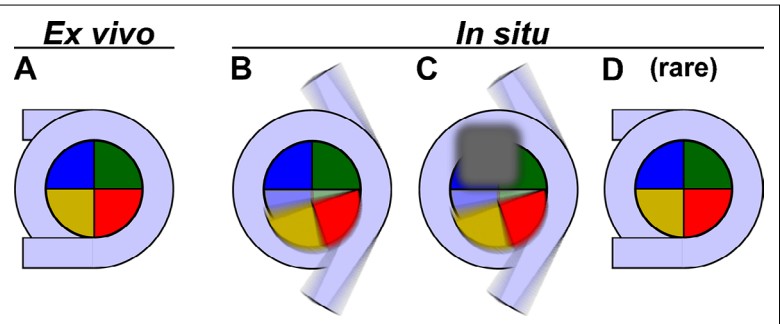

**Figure 5.** Models of yeast nucleosome heterogeneity. Schematics of DNA (light blue) and histones (shaded pie slices) in the nucleosome disc view. The cartoons only illustrate the 147 bp of 'core' DNA. (**A**) Canonical nucleosome, in which all the histones and 147 bp of DNA are part of an ordered complex. (**B**) Nucleosome with alternative histone H2A-H2B (yellow, red) conformations and partially dissociated DNA. (**C**) Nucleosome bound to non-histone proteins (gray). The blurred gray box represents different proteins that can bind, thereby contributing to constitutional heterogeneity. The blurred appearance represents a large range of positions and orientations that protein and DNA components adopt inside cells, which would result in the absence of a class average resembling a canonical nucleosome. (**D**) Canonical nucleosomes are a minority conformation *in situ*.

The online version of this article includes the following figure supplement(s) for figure 5:

**Figure supplement 1.** Simulated tomographic slices of single and stacked nucleosomes.

**Figure supplement 2.** Fourier power spectra analysis of tilt series images.

is attached to the rest of the nucleosome and can have small differences in orientation. The H2A/H2B heterodimers are in contact with the H3/H4 heterotetramer because crosslinking and single-particle fluorescence imaging experiments showed that approximately 80% of H2B – and presumably H2A to which it stably dimerizes – is bound to chromatin *in situ* (*Mohan et al., 2018*; *Ranjan et al., 2020*). H2A-H2B mobility was hinted at by the observations from X-ray crystallography that yeast nucleosomes lack the hydrogen bonds that stabilize the two H2A-H2B heterodimers in metazoan nucleosomes (*White et al., 2001*) and that interfaces between H2A-H2B and H3 are moderately exposed to small-molecule probes (*Marr et al., 2021*). Furthermore, recent simulations show that the H2A/H2B heterodimer may adopt numerous orientations and small displacements, with either fully or partially unwrapped DNA, while remaining in contact with H3-H4 (*Ishida and Kono, 2022*). We also cannot rule out the possibility that expression of H2A-GFP makes nucleosomes less ordered *in situ*.

Like LGY0016 (H2A-GFP sole source), none of the LGY0007 (H3 plus H3-GFP) *in situ* nucleosome-like class averages had an extra density consistent with the GFP density tag (only seen in lysates). Furthermore, there were no canonical nucleosome class averages – either with or without the GFP density – seen among the class averages of LGY0007 cell cryolamella nucleosome-like particles. These observations suggest that the expression of H3-GFP makes nucleosomes less ordered *in situ*. Altogether, our experiments show that in budding yeast, canonical nucleosomes are rare *in situ* and that the expression of GFP-tagged histones leads to further perturbations of nucleosome structure *in situ*.

Our data is consistent with a model in which yeast canonical nucleosomes are abundant *ex vivo* (*Figure 5A*). They adopt multiple non-canonical conformations *in situ* (*Figure 5B and C*) and rarely adopt the canonical one (*Figure 5D*). Note that the blurring in these panels implies the heterogeneity in the positions of the DNA and proteins, not their actual motion. In addition to H2A-H2B and DNA conformational heterogeneity, diverse nucleosome-associated proteins that bind to multiple positions (*Figure 5C*) would further increase the heterogeneity. A high abundance of non-canonical nucleosomes means that more of the genome would be accessible, consistent with yeast having high levels of transcription. Nucleosome heterogeneity is also consistent with the absence of chromatin long-range order *in situ* because crystalline oligonucleosome arrays can only form if sequential nucleosomes adopt nearly identical conformations (*Ekundayo et al., 2017*; *Robinson et al., 2006*; *Song et al., 2014*). Further investigation is needed to identify the biochemical and biophysical factors responsible for the abundance of non-canonical nucleosomes in yeast and to determine their diverse structures (*Zlatanova et al., 2009*).

In combination with our previous work on a HeLa cell (*Cai et al., 2018a*), we show that globular complexes as small as canonical nucleosomes (~200 kDa) can be detected by 3D classification of subtomograms from cryolamellae that were imaged with a VPP. Given the low throughput of cryo-FIB milling, the low yield of cryolamellae thinner than 160 nm, and the technical difficulties of VPP imaging, it is not yet feasible to systematically determine the conditions needed to detect sub-200 kDa complexes by purification *in silico*. More depositions of eukaryote cryolamellae cryo-ET datasets in the public database EMPIAR (*Iudin et al., 2016*) may enable a thorough search of data collection and processing parameter space.

Our *in situ* cryo-ET-based model adds to the body of work on non-canonical nucleosomes. ChIP-seq analysis has detected sub-nucleosomes *in situ*, though the abundance was unknown (*Rhee et al., 2014*). MNase-seq has also detected nucleosomes in a partially unwrapped and partially disassembled state *in situ* (*Ramachandran et al., 2017*). A recent Hi-C variant 'Hi-CO' presented evidence that <147 bp of yeast nucleosomal DNA is protected from MNase attack (*Ohno et al., 2019*), which is consistent with the partial detachment of core DNA. Molecular dynamics advances in all-atom and coarse-grained simulations are now showing that nucleosomes are far more dynamic than previously appreciated (*Armeev et al., 2021*; *Brandani et al., 2021*; *Farr et al., 2021*; *Huertas et al., 2021*; *Ishida and Kono, 2021*). The DNA-unwrapping associated with nucleosome breathing was shown to disfavor ordered helical oligonucleosome structures (*Farr et al., 2021*). Nucleosome breathing is also evident in high-throughput atomic force microscopy experiments, which revealed that only ~30% of nucleosomes are fully wrapped, and that approximately half of the nucleosomes have an opening angle (measured between the entry/exit DNA arms) 60° larger than the fully wrapped one (*Konrad et al., 2021*). Nucleosomes that have non-canonical nucleosome properties, such as lower stability or exposure of internal surfaces, have been reported in fission yeast (*Koyama et al., 2017*; *Sanulli et al., 2019*), which may explain why we did not observe canonical nucleosomes in cryosections in those cells

either (*Cai et al., 2018b*). Some nucleosomes in human and fly cell cryosections appear 'gaping', that is, with the inter-DNA-gyre distance slightly larger than ~2.7 nm (*Eltsov et al., 2018*). Partial DNA detachment has been seen in complexes between nucleosomes and remodelers and transcription factors (*Eustermann et al., 2018*; *Farnung et al., 2017*; *Liu et al., 2017*; *Sundaramoorthy et al., 2018*; *Sundaramoorthy et al., 2017*; *Willhoft et al., 2018*), methyltransferases (*Bilokapic and Halic, 2019*; *Jang et al., 2019*), or transcription-related complexes (*Dodonova et al., 2020*; *Kujirai et al., 2018*). Larger amounts (up to ~25 bp) of DNA detachment have been seen very rarely in cryo-EM structures (*Bilokapic et al., 2018*; *Zhou et al., 2021*). A recent study of cryosectioned fly embryos has also presented evidence of nucleosome-like structures such as hemisomes and three-gyre structures (*Fatmaoui et al., 2022*). The prevalence and functional consequences of non-canonical nucleosomes *in situ* remain to be studied in other organisms.

## Alternative hypotheses

We now consider the alternative hypothesis that canonical nucleosomes are the dominant form *in situ* and that we have missed them as a result of our image processing. Many of the 3D class averages appear to have multiple gyre-like densities, which may arise from nucleosome stacking. However, classes with multiple gyre-like densities are also found in the class averages from the cytoplasm, which does not have nucleosomes (*Figure 1—figure supplement 12*). These are 'junk' classes that result from the averaging of different particle species into the same class. Importantly, stacked canonical nucleosomes are so conspicuous that they could not have been missed in tomographic slices. For instance, two stacked canonical nucleosomes have twice the mass of a single canonical nucleosome and would have dimensions 12 nm by 10 nm (*Figure 5—figure supplement 1*). Multiple stacked canonical nucleosomes would appear like 10-nm-thick filaments.

Another alternative hypothesis is that the lack of disc views, corresponding to nucleosomes whose 'face' is parallel to the cryolamella surface, resulted in canonical nucleosomes going undetected. This hypothesis would be supported only if the orientations of canonical nucleosomes were biased. However, biased orientation in cryo-EM experiments is usually a result of interactions with the air-water interface (*Noble et al., 2018*), which does not exist for nucleosomes inside the nucleus of a cell. Another hypothesis is that the missing views resulted in elongated class averages because of the missing wedge or missing cone in Fourier space. This artifact would only affect the distribution of canonical versus non-canonical nucleosomes if canonical nucleosomes have a preferred orientation, which we have just argued against due to the nature of our cellular samples. The class averages from non-disc-view nucleosomes do not have a missing-wedge/cone artifact because they include nucleosomes whose superhelical axes are in the X-Y plane; when averaged together, the Fourier transforms of these nucleosomes 'fill in' Fourier space because they sample numerous rotations about their super-helical axes. Missing-wedge-free reconstructions are exemplified by plunge-frozen actin filaments, which lie parallel to the EM grid. As illustrated in a recent study (*Merino et al., 2018*), reconstructions of filamentous actin are free of missing-wedge distortions even though the axial views are completely absent.

Some of the particles in the tomographic slices resemble donuts. Because nucleosome disc views also resemble donuts, it is possible that we completely missed these particles in either our template matching or classification analyses. As stated above, even if we missed all the disc views, our conclusions would not change because the orientations of canonical nucleosomes inside of cellular samples are not biased. Furthermore, the donut-like particles cannot be canonical nucleosomes because they are too wide (>12 nm). A subset of them were detected by template matching and classification and separated into their own class, which resembles a cylinder with a rounded cap (*Figure 4—video 1*, row 5 column 1).

Another hypothesis for the low numbers of detected canonical nucleosomes is that the nucleoplasm is too crowded, making the image processing infeasible. However, crowding is an unlikely technical limitation because we were able to detect canonical nucleosome class averages in our most-crowded nuclear lysates, which are so crowded that most nucleosomes are butted against others (*Figure 3—figure supplements 3 and 4*). Crowding may instead have biological contributions to the different subtomogram analysis outcomes in cell nuclei and nuclear lysates. For example, the crowding from other nuclear constituents (proteins, RNAs, polysaccharides, etc.) may contribute to *in situ* nucleosome structure, but is lost during nucleus isolation.

## Limitations of the study

Because naked linker DNA cannot yet be seen *in situ* in yeast, the connectivity between the nucleosomes (both canonical and non-canonical) could not be followed. The subtomogram analysis was unable to resolve the structures of the non-canonical nucleosomes. Because of the low resolution, we could not assess the variability in the structure and composition of the canonical nucleosomes. All of these limitations could potentially be addressed by increasing the data quality and the numbers of classes and subtomograms per class, which require advances at all stages of structural cell biology. These advances include improvements in cryo-FIB milling throughput and reproducibility (*Buckley et al., 2020*; *Tacke et al., 2021*; *Zachs et al., 2020*), cryo-EM cameras, laser phase plates (*Turnbaugh et al., 2021*), template-matching/segmentation software (*Bepler et al., 2020*; *Moebel et al., 2021*), and subtilt refinement software such as emClarity (*Himes and Zhang, 2018*), EMAN2 (*Chen et al., 2019*), Warp/M (*Tegunov et al., 2021*), and RELION 4 (*Zivanov et al., 2022*). Because structural cell biology is still a nascent field, the optimum combination of sample prep, imaging, and data analysis will require thorough exploration of the parameter space at each step. The nucleosome VPP subtomogram averages presented here were limited to ~24 Å, even though the data was recorded close to focus as suggested by a higher-resolution study of purified ribosomes (*Khoshouei et al., 2017*). Limiting factors include lower particle numbers, alignment inaccuracy, VPP charging, and the lack of contrast transfer function (CTF) modeling. In single-particle cryo-EM analysis, high-resolution analysis (3 Å or better) requires accurate modeling and then compensation for the CTF. A key first step of CTF modeling is the visualization of Thon rings in Fourier power spectra. We plotted Fourier power spectra in both 2D with CTFFIND (*Rohou and Grigorieff, 2015*) and as a 1D rotational average with IMOD (*Mastronarde, 1997*), but could not see Thon rings (*Figure 5—figure supplement 2*), meaning that CTF compensation is not feasible. A study of VPP single-particle cryo-EM analysis recommends intentionally underfocusing to 500 nm to increase the number of Thon rings, making CTF modeling easier (*Danev et al., 2017*). However, the samples used in that study were much thinner (plunge-frozen proteasomes) and the dose used per image was much higher (~40 electrons/Å$^2$). Because the subtomogram analysis of smaller complexes inside cryolamellae faces multiple challenges, more investigation is needed to determine the optimum parameters for increased resolution.

Another challenge is the positive identification of the non-canonical nucleosome species. The approaches attempted here (the use of a GFP tag) failed. An alternative approach would be to remap the various averages and show that the linker DNA of sequential non-canonical nucleosomes point to each other, as we have previously done for HeLa chromatin *in situ* (*Cai et al., 2018a*). Unfortunately, this approach is not suitable for yeast because only the canonical class averages have linker-DNA densities – none of the other class averages have densities that resemble linker DNA. Because non-canonical yeast nucleosomes *in situ* are unknown structures, we would have to identify individual instances using a 'GFP of cryo-EM', in the form of a compact fusion protein-like structure that can be directly visualized in tomographic slices without classification and subtomogram averaging. Several attempts have been made in recent decades (*Diestra et al., 2009*; *Mercogliano and DeRosier, 2007*; *Nishino et al., 2007*; *Wang et al., 2011*). More work is needed to test the suitability of such tags as 'GFPs of cryo-EM' *in situ*.

## Materials and methods

Key resources table is in Appendix 1.

### Yeast strain and growth conditions

All yeast strains were streaked on yeast extract peptone dextrose (YPD) agar plates (2% wt/vol peptone, 1% wt/vol yeast extract, 2% wt/vol glucose for liquid media; additional 2% wt/vol agar for agar plates) at 30°C and cultured in YPD in conical flasks shaking at 200–250 RPM at 30°C. All yeast strains are of mating type a. Modifications or lack thereof to all histone genes were authenticated by PCR and Sanger sequencing (Bio Basic Asia Pacific Pte Ltd, Singapore) in all yeast strains.

### Bacterial growth conditions

All plasmids were provided in DH5-Alpha *Escherichia coli*. The bacteria were streaked on LB agar plates with ampicillin (40 g/L LB Broth with agar [Miller], 100 µg/mL ampicillin) at 37°C and cultured

in LB liquid medium with ampicillin (25 g/L LB Broth [Miller], 100 μg/mL ampicillin) in vent cap tubes shaking at 200–250 RPM at 37°C.

## Plasmid extraction and linearization

The plasmid pFA6a-GFP(S65T)-His3MX6 (*Longtine et al., 1998*) was a gift from John Pringle (Addgene 41598; Addgene, Watertown, MA, USA) and pFA6a-link-yoTagRFP-T-Kan (*Lee et al., 2013*) was a gift from Wendell Lim & Kurt Thorn (Addgene 44906); both were given in the form of bacterial stabs. Note that pFA6a-link-yoTagRFP-T-Kan is the source of the KanR marker used to delete HTB2-HTA2. Five mL of bacteria were cultured overnight, then plasmids were extracted with the QIAprep Spin Miniprep Kit (QIAGEN, Hilden, Germany) following the manufacturer's instructions.

Extracted plasmids were linearized by double digestion with a reaction containing 1 μg of plasmid DNA, 5 μL of 10× rCutSmart Buffer (New England BioLabs, Ipswich, MA, USA), 10 units each of *Sal*I and *EcoR*V restriction enzymes (New England BioLabs) topped up to 50 μL with nuclease-free water. The reaction mixture was heated at 37°C for 15 min for the digestion reaction, then 80°C for 20 min to inactivate the enzymes.

## Strain construction

The strain details are shown in *Supplementary file 1*. Primers were from IDT (Integrated DNA Technologies, Inc, Singapore) and listed in *Supplementary file 2*. Q5 PCR Master Mix (New England BioLabs) was used for PCRs. BY4741 (*Brachmann et al., 1998*) served as the wild-type strain. Tagging and deletion cassettes were amplified from linearized pFA6a plasmids (*Lee et al., 2013*; *Longtine et al., 1998*) with a PCR containing 1 ng of template DNA and 0.5 μM of each primer, using a PCR program of 98°C for 30 s, 30 cycles of 98°C for 5 s, 60°C for 10 s and 72°C for 1.5 min, then 72°C for 5 min.

Cells were transformed using the lithium acetate/PEG4000 method reported in *Nishimura and Kanemaki, 2014*. Overnight cell culture in YPD was diluted to 0.1 $OD_{600}$ in 25 mL of YPD and grown to 0.3 $OD_{600}$. Ten mL of cells were collected, centrifuged at 1600×$g$ at 25°C for 3 min, and the supernatant was removed. The cells were then washed twice with 10 mL sterile water with centrifugation at 1600×$g$ at 25°C for 3 min. The cells were resuspended in 1 mL sterile water, transferred to a new 1.5 mL collection tube, centrifuged at 17,900×$g$ at 25°C for 1 min, washed in 1 mL of TE/LiAc (10 mM Tris, 1 mM ethylenediaminetetraacetic acid [EDTA], 100 mM lithium acetate), with centrifugation at 17,900×$g$ for 1 min, and resuspended in 50 μL of the same buffer. Fifty μL of cell suspension was transferred to a new 1.5 mL collection tube containing 5 μL of 10 mg/mL salmon sperm DNA (Sigma-Aldrich, Burlington, MA, USA) plus 5 μL of PCR-amplified cassette DNA. 360 μL of TE/LiAc/PEG (10 mM Tris, 1 mM EDTA, 100 mM lithium acetate, 40% wt/vol polyethylene glycol 4000) was added and incubated with shaking at 25°C for 30 min. Forty μL of dimethyl sulfoxide was added, the suspension was incubated at 42°C for 15 min in a water bath, then cooled on ice for 2 min. The suspension was centrifuged at 13,000×$g$ for 1 min, the liquid was removed, and the cells were resuspended in 300 μL of 1× TE buffer, pH 8.0 (10 mM Tris, 1 mM EDTA). Two hundred μL of this cell suspension was plated on a selection plate and incubated for several days at 30°C. Histidine selection plates were created with 6.7 g/L yeast nitrogen base without amino acids (Sigma-Aldrich), 1.92 g/L yeast synthetic drop-out medium supplements without histidine (Sigma-Aldrich), 2% wt/vol glucose, and 2% wt/vol agar. G418 selection plates were created by adding G418 to the molten agar (YPD for G418 single selection, histidine auxotrophy medium for double selection) to a concentration of 200 mg/L before it was poured.

Transformants were verified by PCR and Sanger sequencing. Genomic DNA was extracted with the DNeasy Blood & Tissue Kit (QIAGEN) following the manufacturer's instructions. Confirmation PCR was performed with 1 μg of template genomic DNA and 0.5 μM of each primer, using a PCR program of 94°C for 2 min, 30 cycles of 94°C for 1 min, 62°C for 1 min, and 72°C for 3.5 min, then 72°C for 5 min. PCR products were purified with the QIAquick PCR Purification Kit (QIAGEN) following the manufacturer's instructions, with water used for the final elution before they were sent for Sanger sequencing.

## DNA gels and immunoblots

PCR products were electrophoresed in 2% agarose in Tris-acetate-EDTA and visualized with Floro-Safe DNA Stain (Axil Scientific Pte Ltd, Singapore). The electrophoresis was performed at 100 V for 60–80 min before visualization with a G:Box (Syngene).

To generate protein samples for immunoblot analysis, ~20 $OD_{600}$ units of yeast cells were pelleted and stored at −80°C for at least 1 hr. Cells were then resuspended in 200 µL of ice-cold 20% trichloroacetic acid (TCA) and vortexed with glass beads at 4°C for 1 min four times. For each sample, 500 µL of ice-cold 5% TCA was added, mixed with the pellet, then transferred to a new 1.5 mL collection tube. Another 500 µL of ice-cold 5% TCA was mixed with each pellet and transferred to the same 1.5 mL collection tube as before, so that each collection tube had 1 mL total volume. The tubes were then left on ice for 10 minutes and centrifuged at 15,000×*g* at 4°C for 20 min. The TCA was aspirated from the tube, then the pellet was resuspended in 212 µL of Laemmli sample buffer with 2-mercaptoethanol added (Bio-Rad, Hercules, CA, USA). To neutralize the residual TCA, 26 µL of 1 M Tris, pH 8 was added. This mixture was heated at 95°C for 5 min, centrifuged at 25°C at 15,000×*g* for 10 min, then 5 µL of the supernatant was subjected to SDS-PAGE.

The primary and secondary antibodies used for immunoblots are shown in *Supplementary file 3*. Proteins were electrophoresed in 10% Mini-PROTEAN TGX Precast Protein Gels (Bio-Rad). Either Precision Plus Protein WesternC Standards (Bio-Rad) or Invitrogen MagicMark XP Western Protein Standard (Thermo Fisher Scientific, TFS, Waltham, MA, USA) served as the ladder. The proteins were electrophoresed at 100 V for 1 hr at 25°C. The gels were transferred onto Immun-Blot PVDF Membranes (Bio-Rad) at 4°C in transfer buffer (3.02 g/L Tris, 14.4 g/L glycine, 20% methanol). The transfer was performed at 100 V for 30 min. The membranes were then blocked with 2% BSA in 1× Tris-Buffered Saline, 0.1% Tween 20 Detergent (TBST) for 1 hr. This was followed by 50 µg/mL avidin in TBST for 30 min, then a wash with TBST for 20 min if Precision Plus Protein WesternC Standards ladder was used. All antibody dilution factors are reported in *Supplementary file 3*. The membranes were probed with primary antibodies at the stated dilutions in 2% BSA in TBST for 1 hr at 25°C. Membranes were washed with TBST for 20 min three times at 25°C. The membranes were then probed with secondary antibodies in 2% BSA in TBST for 1 hr, with additional Precision Protein StrepTactin-HRP Conjugate (Bio-Rad) at 1:10,000 dilution if Precision Plus Protein WesternC Standards ladder was used. Finally, the membranes were washed with TBST for 10 min three times, then treated with a 50:50 mixture of Clarity Western Peroxide Reagent and Clarity Western Luminol/Enhancer Reagent (Bio-Rad) for 5 min before visualization by chemiluminescence on an ImageQuant LAS 4000 (Cytiva, Marlborough, MA, USA).

## Fluorescence microscopy

Cells were grown to log phase ($OD_{600}$=0.1–1.0), of which 2 $OD_{600}$ units of cells were collected, pelleted at 5000×*g* for 1 min, then resuspended in 1 mL of YPD. Four µL of cell culture was then applied to a glass slide and pressed against a coverslip. The cells were imaged live at 23°C with an Olympus FV3000 Confocal Laser Scanning Microscope (Olympus, Tokyo, Japan) equipped with a 1.35 NA 60× oil-immersion objective lens. GFP fluorescence was acquired using the 488 nm laser line, with a DIC image recorded in parallel with the fluorescence image. Images were captured as Z-stacks thick enough to sample the GFP signals through all the nuclei in each stage position. Additional details regarding data collection are shown in *Supplementary file 4*.

## Preparation of nuclear lysates

Yeast nuclei were prepared with reagents from the Yeast Nuclei Isolation Kit (Abcam 206997, Cambridge, UK), unless noted otherwise. Yeast cells (30 mL, $OD_{600}$~1) were pelleted at 3000×*g* for 5 min at 25°C. The pellet was washed twice with 1 mL water (3000×*g*, 1 min). The pellet was then resuspended in 1 mL Buffer A (pre-warmed to 30°C) containing 10 mM dithiothreitol. The suspension was incubated in a 30°C water bath for 10 min. Cells were then pelleted at 1500×*g* for 5 min and then resuspended in 1 mL Buffer B (pre-warmed to 30°C) containing lysis enzyme cocktail (1:100 dilution). The suspension was incubated in a 30°C shaker for 15 min for cell wall digestion and then pelleted at 1500×*g* for 5 min at 4°C. The pellet was resuspended in 1 mL pre-chilled Buffer C with protease inhibitor cocktail (1:1000 dilution). The cells were lysed by 15 up-and-down strokes with a pre-chilled glass Dounce homogenizer on ice. The lysate was incubated with shaking for 30 min at 25°C. The

cell debris was pelleted at 500×$g$ for 5 min at 4°C. The supernatant was then transferred to a new tube. The nuclei were pelleted at 20,000×$g$ for 10 min at 4°C. The nuclear pellet was resuspended in 10–20 µL pre-chilled lysis buffer (50 mM EDTA and 1:1000 protease inhibitor cocktail dilution) and incubated for 15 min on ice.

Nuclei lysates (3 µL) were added to a glow-discharged CF-4/2-2C-T grid (Protochips, Morrisville, NC, USA). The grid was plunge-frozen using a Vitrobot Mark IV (blot time: 1 s, blot force: 1, humidity: 100%, temperature: 4°C).

## Preparation of cryosections

Self-pressurized freezing was done based on a modified version of a published protocol (*Yakovlev and Downing, 2011*). Yeast cells (30 mL, OD$_{600}$=0.2–0.6) were pelleted and resuspended in a dextran stock (40 kDa, 60% wt/vol, in YPD) to a final concentration of 30%. Cells were then loaded into a copper tube (0.45/0.3 mm outer/inner diameters). Both ends of the tube were sealed with flat-jaw pliers. The tube was held horizontally and dropped into the liquid-ethane cryogen. The tube's ends were removed under liquid nitrogen with a tube-cut tool (Engineering Office M. Wohlwend, Sennwald, Switzerland).

Gold colloid solution (10 nm diameter, 5 µL at 5.7×10$^{12}$ particles/mL in 0.1 mg/mL BSA) was applied to a continuous-carbon grid (10-nm-thick carbon) and then air-dried overnight. Cryosections were controlled by a custom joystick-based micromanipulator (MN-151S, Narishige Co., Ltd., Tokyo, Japan) (*Ladinsky et al., 2006*; *Ng et al., 2020*). Seventy-nm-thick frozen-hydrated sections were cut at −150°C in a Leica UC7/FC7 cryo-ultramicrotome (Leica Microsystems, Vienna, Austria). The EM grid was positioned underneath the ribbon using a Leica micromanipulator (*Studer et al., 2014*). The cryo-section ribbon (~3 mm long) was then attached to the grid by operating the Crion (Leica Microsystems) in 'charge' mode for ~30 s (*Pierson et al., 2010*).

## Preparation of cryolamellae

Cells were plunge-frozen and then cryo-FIB milled using the method of *Medeiros et al., 2018*, as follows. Immediately before plunge-freezing, mid-log phase (OD$_{600}$~0.6) yeast cells were pelleted at 4000×$g$ for 5 min. They were then resuspended in YPD media containing 3% (vol/vol) dimethyl sulfoxide as cryo-protectant to a final OD$_{600}$ of approximately 2.5. Four µL of the cells were subsequently deposited onto Quantifoil R2/4 200 mesh copper grids (Quantifoil Micro Tools GmbH, Jena, Germany), which were then manually blotted from the back with Whatman Grade 1 filter paper for approximately 3–5 s. The grids were then plunged into a 63/37 propane/ethane mixture (*Tivol et al., 2008*) using a Vitrobot Mark IV (humidity: 100%, temperature: 4°C). Cryo-FIB milling was performed on a Helios NanoLab 600 DualBeam (Thermo Fisher Scientific, TFS, Waltham, MA, USA) equipped with a Quorum PolarPrep 2000 transfer system (Quorum Technologies, Laughton, UK). Plunge-frozen yeast samples were coated with a layer of organometallic platinum using the in-chamber gas injection system and the cold deposition method (*Hayles et al., 2007*). Cryolamellae were then generated as follows: bulk material was first removed using the FIB at 30 kV 2.8 nA, followed by successive thinning of the cryolamellae at lower currents of 0.28 nA and 48 pA.

## Cryo-ET data collection and reconstruction

All cryo-ET data were collected on Titan Krioses (TFS). Tilt series were collected with either TFS Tomo4, Leginon (*Suloway et al., 2009*), SerialEM (*Mastronarde, 2003*), or PACE-tomo (*Eisenstein et al., 2023*). Images were recorded either on a Falcon II (TFS) in integration mode or as movie frames on a K2 or K3 summit camera (Gatan, Pleasanton, CA, USA) in super-resolution mode. The pixel sizes for the K2 and K3 data were chosen so that when binned to the same level, they closely match the ~7 Å pixel size in our previous *in situ* study of HeLa chromatin (*Cai et al., 2018a*). Smaller pixel sizes were chosen for Falcon II data because this camera has lower detective quantum efficiency than the K-series cameras (*Ruskin et al., 2013*). Movies were aligned with either MotionCor2 (*Zheng et al., 2017*) or IMOD alignframes (*Mastronarde, 1997*). Prior to starting data collection of cryolamellae, the stage was pre-tilted to either −10° or −15° to account for the milling angle. Additional data collection details are shown in *Supplementary file 5*.

VPP imaging was done using the protocol of *Fukuda et al., 2015*. The VPP position and condenser stigmator were adjusted such that the Ronchigram (the projected pattern visible in the camera) was

free of elongated features, indicating no astigmatism, and is flat rather than grainy, indicating that the VPP is on-plane. Notably, the VPP needs to be heated, or else the phase shift will rapidly exceed π/2 radians (*Danev et al., 2014*). Two different VPP assemblies were used, which we will refer to as the NYSBC-2019 and NUS-2020 ones. These VPPs required different heater power settings to maintain a stable phase shift. The NYSBC-2019 and NUS-2020 VPP heaters were operated at ~100 mW and ~370 mW, respectively.

Cryotomograms were reconstructed using IMOD's *eTomo* workflow (*Mastronarde, 1997*). Tilt series of lysates and cryosections were aligned using the gold beads as fiducials while those of cryolamellae were aligned with patches as fiducials. Only the tilt series that exhibited the minimal amount of drift and sample warping were analyzed. To further improve the tilt series alignment around the chromatin, fiducials (beads or patches) were chosen on the chromatin regions. For cryolamella patch tracking, tilt series were binned to a pixel size of 6.8 Å, the 'Break contours into pieces w/ overlap' was set to 10, and the low-frequency rolloff sigma and cutoff radius were set to 0.03 and 0.1 pixel$^{-1}$, respectively. A boundary model was created so that only the chromatin was enclosed. Fiducial patches were manually deleted if they overlapped with debris-like ice crystals or if they mistracked. We did not detect a correlation between the alignment residual (*Supplementary file 6*) and our ability to detect canonical nucleosomes.

CTF estimation and phase flipping were done on the defocus phase-contrast datasets using IMOD's *ctfplotter* and *ctfphaseflip* programs. Prior to reconstruction, the cryosection and cryolamellae tilt series were binned to a final pixel size of 6.8 Å using the *eTomo* antialiasing option. Two tomogram versions were reconstructed for each tilt series. For visualization purposes, the tilt series were low-pass filtered to attenuate spatial frequencies beyond 25 Å to 30 Å resolution, prior to tomogram reconstruction. For classification analysis, the tilt series were low-pass filtered with a Gaussian rolloff starting at 15 Å resolution for lysates and 20 Å for cryosections and cryolamellae. In the classification jobs, the resolution of the data was further limited to 25 Å or 20 Å (see next section for details). More details of the datasets analyzed in this paper are shown in *Supplementary file 6*.

## Nucleosome template matching, classification, and subtomogram averaging

A featureless round-edged 10-nm-diameter, 6-nm-thick cylindrical template was created using the Bsoft program *beditimg* (*Heymann and Belnap, 2007*). A cubic search grid with a 12 nm spacing was created with the PEET program *gridInit* (*Heumann, 2016*). Regions that had high-contrast artifacts from surface contaminants and ice crystals were excluded. Template matching was done using PEET (*Heumann, 2016*; *Heumann et al., 2011*; *Nicastro et al., 2006*), with a duplicate removal cutoff distance of 6 nm. To accelerate the runs, no orientation search was done around the cylindrical axis and the resolution was attenuated starting at 70 Å on account of the smooth appearance of the template. Candidate hit lists of different cross-correlation cutoffs were generated using the PEET program *createAlignedModel*, then visualized together with the tomograms in *3dmod*. The cross-correlation cutoff that eliminated spurious densities (primarily empty nucleoplasm) was chosen. The final numbers of subtomograms analyzed for each sample are in *Supplementary file 7* and in the figures and figure legends.

Classification and subtomogram analysis were done with RELION (*Kimanius et al., 2016*; *Scheres, 2012*), following the workflows in *Figure 1—figure supplement 1*. In the published workflow (*Figure 1—figure supplement 1*, panel A) (*Bharat and Scheres, 2016*), each subtomogram is first averaged by projecting the entire volume (16 nm along the Z axis), which introduces contributions from densities above and below the candidate nucleosome. To minimize the influence of other densities, pseudo-projections were created by averaging ~12 nm along the Z axis, using the *ot_relion_project.py* script. 2D classification using mask diameters ranging from 120 Å to 140 Å produced clear nucleosome-like classes, though the smaller masks included fewer adjacent densities. Note that in RELION, only circular masks are available for 2D classification, meaning that it is not possible for the pseudo-projected densities to appear cylinder-like due to truncation by the mask. Densities that belonged to the most nucleosome-like classes were exported for 3D classification, split into 30 classes. The resolution cutoffs were 25 Å for 2D and 20 Å for 3D classification. To eliminate the influence of adjacent densities during 3D classification, a smooth cylindrical mask with a cosine-shaped edge was applied. The mask was created using *beditimg* and *relion_mask_create*. Because the GFP densities

protrude from the nucleosome surface, we used a 9-nm-tall cylindrical mask for the analysis of nucleosomes with GFP tags, such as LGY0016 nucleosomes. The percentage of subtomograms belonging to each class was extracted with the script *count_particles.awk* from *Gaullier, 2021*.

As observed in our previous study (*Cai et al., 2018a*), some canonical nucleosomes were lost in the 2D classification process. We therefore used the alternative workflow in which the template matching hits were directly classified in 3D (*Figure 1—figure supplement 1*, panel B). To accommodate the increased diversity of complexes (some of which would have been removed had 2D classification been done), we used 40 classes for BY4741, LGY0015, and LGY0016 nuclear lysates and 100 classes for cell cryolamellae and nuclear lysates of strains in which H3 or H4 were GFP tagged, without prior 2D classification. These jobs crashed frequently because the RELION memory usage scales up with the number of classes (*Kimanius et al., 2016*). We were able to eliminate the crash problem by using higher-memory GPUs and by decreasing the number of translational search steps from 5 to 3 or 4. The canonical nucleosome classes were subjected to 'gold-standard' 3D refinement (*Henderson et al., 2012*). No map sharpening was applied. Subtomogram class average volumes were visualized with UCSF Chimera (*Pettersen et al., 2004*).

To visualize the distribution of the canonical nucleosome class averages in the cryolamellae, they were remapped into their positions in the tomogram using the *ot_remap.py* script.

## Biased reference classification control

A 14 Å resolution density map was simulated from the yeast nucleosome crystal structure PDB 1ID3 (*White et al., 2001*) using the Bsoft program *bgex*. To account for the artifacts associated with defocus phase contrast and CTF correction, a 6 μm underfocus was applied and then 'corrected' for using the Bsoft program *bctf*. The map was also subjected to the 20 Å resolution low-pass filter that was used on the tilt series using the IMOD program *mtffilter*. Template matching was done using this reference, including data to higher resolution (28 Å) than that for the less-biased search above and including a search around all Euler angles. The hits were subjected to 2D classification to remove obvious non-nucleosomal densities. Next, the hits were 3D classified using the simulated map as an initial alignment reference. To maximize the model bias, the template was only low-pass filtered to 20 Å resolution instead of the recommended 60 Å (*Bharat and Scheres, 2016*).

## Estimation of nucleosomes sampled per cell cryolamella

First, the average concentration of nucleosomes in the chromatin was estimated. The absolute number of nucleosomes per cell determined from genomics is 60,000 (*Oberbeckmann et al., 2019*). Soft X-ray tomography measurements revealed that the average G1 nucleus volume is 2 μm$^3$, of which 20% is nucleolus (*Uchida et al., 2011*). Accordingly, chromatin (the nuclear volume not taken by the nucleolus), which contains the vast majority of nucleosomes, occupies ~1.6 μm$^3$. These two experimental values give an average nucleosome concentration of 37,500 per μm$^3$. Next, the nuclear volume sampled by subtomogram analysis was determined by first drawing one closed contour around the chromatin using 3dmod; this closed contour encloses only the portion of the tomogram that was analyzed by template matching. The volume of this closed contour, which is one-voxel thick, was extracted using the command: imodinfo -F model.mod.

This command outputs the quantity 'Cylinder volume', in cubic pixels (voxels). The total tomographic volume sampled (*Supplementary file 8*) was obtained by multiplying Cylinder volume, the voxel volume (0.31 nm$^3$ for 0.68 nm pixel size), and the number of tomographic slices that contain chromatin. The BY4741 VPP tomograms summed to 0.67 μm$^3$, which yields ~25,000 nucleosomes. For the HeLa cell in *Cai et al., 2018a* (EMPIAR-10179), the nucleus volume analyzed was 0.12 μm$^3$.

## Simulations of nucleosome tomographic slices

Atomic models of the nucleosome (PDB 1KX5) (*Davey et al., 2002*) were manually positioned in UCSF Chimera and edited to remove the N-terminal tails. A 3D density map was calculated with the Bsoft program *bgex*. There is no software that simulates Volta contrast, so we approximated the Volta-induced phase shift by setting the amplitude contrast to 100% and the defocus to zero in the Bsoft program *bctf*, followed by 'correction', also done with *bctf*. A tilt series was calculated from the simulated map using the IMOD program *xyzproj*. The tilt series was then aligned by cross-correlation and then back-projected using IMOD's *eTomo* workflow. Parameters such as pixel size, tilt range, and

tilt angle were kept as close to the experimental ones as possible. Tomographic slices were made in IMOD slicer at the same thickness as for the real cryotomograms.

### Ribosome subtomogram analysis control

Ribosomes were analyzed using the same software packages as for nucleosomes, as detailed in the workflow in *Figure 4—figure supplement 6*. Candidate ribosomes were template matched in a single VPP tomogram (from *Figure 1—figure supplement 11*), using a 25-nm-diameter sphere as a reference. The candidate ribosomes were filtered by cross-correlation so that obvious false positives (vacuum) were excluded, leaving 3816 hits. These particles were subjected to direct 3D classification with k=10, resulting in six non-empty classes. Particles belonging to the two ribosome classes (1150 total) were pooled and 'gold-standard' refined, yielding a density map at 28 Å resolution (FSC = 0.143)/33 Å resolution (FSC = 0.5). Density maps were simulated at 15 Å, 20 Å, and 30 Å resolution using the Bsoft program *bgex* and the yeast ribosome crystal structure (*Ben-Shem et al., 2011*). To approximate the use of the VPP, the amplitude contrast was set to 90% and the defocus to −1 μm.

### Materials, data, and code availability

All *S. cerevisiae* strains generated in this study are available upon request. A subtomogram average of a BY4741 canonical nucleosome *ex vivo*, a double-GFP tagged LGY0016 nucleosome *ex vivo*, and the two BY4741 canonical nucleosome classes *in situ* have been deposited at EMDB as entry EMD-31086. All raw cryo-ET data, reconstructed tomograms, and BY4741 cryolamellae VPP *in situ* class averages have been deposited in EMPIAR under entry EMPIAR-10678. All auxiliary scripts have been deposited at GitHub (https://github.com/anaphaze/ot-tools, *Gan, 2019*; copy archived at swh:1:rev:91a336987046066ffc9fb0ef9256b3a72513b92b) and are publicly available as of the date of publication. Any additional information required to reanalyze the data reported in this paper is available upon request.

## Acknowledgements

We thank Kerry Bloom, Vu Nguyen, and Carl Wu for advice on fluorescent protein tagging of histones; Bill Rice and Ed Eng for help with cryo-EM data collection; John Heumann for discussions about PEET and for implementing a parallelized duplicate removal routine; Rado Danev, Kliment Verba, and Shenping Wu for advice on the VPP. The Quadro P6000 used in this work was kindly donated by the NVIDIA Corporation. Some of this work was performed at the Simons Electron Microscopy Center and National Resource for Automated Molecular Microscopy located at the New York Structural Biology Center.

## Additional information

### Funding

| Funder | Grant reference number | Author |
| --- | --- | --- |
| Ministry of Education - Singapore | R-154-000-A49-114 | Lu Gan<br>Zhi Yang Tan<br>Shujun Cai<br>Jon K Chen |
| Ministry of Education - Singapore | MOE2019-T2-1-140 | Lu Gan<br>Zhi Yang Tan<br>Shujun Cai<br>Jon K Chen |
| National Institute of General Medical Sciences | F32GM128303 | Alex J Noble |
| Simons Foundation | SF349247 | Alex J Noble |

| Funder | Grant reference number | Author |
| --- | --- | --- |
| Empire State Development's Division of Science, Technology and Innovation | NYSTAR | Alex J Noble |
| National Institute of General Medical Sciences | GM103310 | Alex J Noble |
| National Institutes of Health | S10 RR029300 | Alex J Noble |

The funders had no role in study design, data collection and interpretation, or the decision to submit the work for publication.

## Author contributions

Zhi Yang Tan, Shujun Cai, Alex J Noble, Jon K Chen, Investigation, Methodology, Writing - original draft, Writing - review and editing; Jian Shi, Methodology; Lu Gan, Conceptualization, Resources, Data curation, Software, Formal analysis, Supervision, Funding acquisition, Validation, Investigation, Visualization, Methodology, Writing - original draft, Project administration, Writing - review and editing

### Author ORCIDs
Zhi Yang Tan ⓘ http://orcid.org/0000-0002-5297-4427
Alex J Noble ⓘ http://orcid.org/0000-0001-8634-2279
Jon K Chen ⓘ http://orcid.org/0000-0002-3837-5444
Lu Gan ⓘ http://orcid.org/0000-0002-8685-4896

Reviewer #1 (Public Review): https://doi.org/10.7554/eLife.87672.3.sa1
Reviewer #2 (Public Review): https://doi.org/10.7554/eLife.87672.3.sa2
Reviewer #3 (Public Review): https://doi.org/10.7554/eLife.87672.3.sa3
Author Response https://doi.org/10.7554/eLife.87672.3.sa4

# Additional files

## Supplementary files
- Supplementary file 1. Table of genotypes of strains used in this paper.
- Supplementary file 2. Table of PCR primers. All primers are listed in the direction 5' → 3'.
- Supplementary file 3. Table of antibodies for immunoblots. CST = Cell Signaling Technology.
- Supplementary file 4. Table of confocal microscopy details.
- Supplementary file 5. Table of cryo-electron tomography (cryo-ET) details.
- Supplementary file 6. Table of cryotomogram details. All data reported in this table were used for subtomogram analysis and were deposited as EMPIAR-10678. The K2 and K3 raw data were collected in super-resolution mode, with ½ the pixel size reported in the table. Pixel size therefore refers to the camera's 'bin ×1' pixel. * Refined defocus (Δf) values are reported for defocus phase-contrast data while nominal defoci are reported for Volta phase-contrast (VPP) data. Δtilt = tilt increment. Figure = figures that show this dataset; those without a figure number were used for subtomogram averaging. Δtilt = tilt increment. Camera: FII = Falcon II, K2 = K2 GIF, K3 = K3 GIF. Thickness is measured from the reconstructed tomogram. For cryosections, the thickness is variable due to the presence of crevasses. Residual = alignment residual, in nanometers.
- Supplementary file 7. Table of total nucleosome-like particles analyzed.
- Supplementary file 8. Table of nucleus volume sampled for subtomogram analysis.
- MDAR checklist

## Data availability
A subtomogram average of a BY4741 canonical nucleosome *ex vivo*, a double-GFP tagged LGY0016 nucleosome *ex vivo*, and the two BY4741 canonical nucleosome classes *in situ* have been deposited at EMDB as entry EMD-31086. All cryo-ET raw data, reconstructed tomograms and BY4741 cryolamellae VPP *in situ* class averages have been deposited in EMPIAR under entry EMPIAR-10678.

The following datasets were generated:

| Author(s) | Year | Dataset title | Dataset URL | Database and Identifier |
|---|---|---|---|---|
| Tan ZY, Cai S, Noble AJ, Chen JK, Shi J, Gan L | 2022 | Subtomogram average of a yeast nucleosome from BY4741 cells | https://www.ebi.ac.uk/emdb/EMD-31086 | Electron Microscopy Data Bank, EMD-31086 |
| Tan ZY, Cai S, Noble AJ, Chen JK, Shi J, Gan L | 2023 | Heterogeneous non-canonical nucleosomes predominate in yeast cells *in situ* | https://www.ebi.ac.uk/empiar/EMPIAR-10678 | Electron Microscopy Public Image Archive, EMPIAR-10678 |

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

# Appendix 1

## Appendix 1—key resources table

| Reagent type (species) or resource | Designation | Source or reference | Identifiers | Additional information |
|---|---|---|---|---|
| Gene (*Saccharomyces cerevisiae*) | HTA1 | Saccharomyces Genome Database | SGD:S000002633 | |
| Gene (*S. cerevisiae*) | HTB1 | Saccharomyces Genome Database | SGD:S000002632 | |
| Gene (*S. cerevisiae*) | HTA2 | Saccharomyces Genome Database | SGD:S000000099 | |
| Gene (*S. cerevisiae*) | HTB2 | Saccharomyces Genome Database | SGD:S000000098 | |
| Gene (*S. cerevisiae*) | HHT1 | Saccharomyces Genome Database | SGD:S000000214 | |
| Gene (*S. cerevisiae*) | HHF1 | Saccharomyces Genome Database | SGD:S000000213 | |
| Strain, strain base (*S. cerevisiae*) | BY4741 | EUROSCARF | Y00000 | MATa his3D1 leu2Δ0 met15Δ0 ura3Δ0, parent strain for transformation |
| Cell line (*S. cerevisiae*) | LGY0012 | This paper | | (hta2-htb2)Δ0::KANMX, derived from BY4741 |
| Cell line (*S. cerevisiae*) | LGY0015 | This paper | | HTB1-GFP(S65T)(0aa linker)-HIS3MX, derived from BY4741 |
| Cell line (*S. cerevisiae*) | LGY0016 | This paper | | (hta2-htb2)Δ0::KANMX HTA1-GFP(S65T) (0aa linker)-HIS3MX, derived from LGY0012 |
| Cell line (*S. cerevisiae*) | LGY0002 | This paper | | HHT1-GFP(S65T)(RIPGLIN linker)-HIS3MX, derived from BY4741 |
| Cell line (*S. cerevisiae*) | LGY0007 | This paper | | HHT1-GFP(S65T)(0aa linker)-HIS3MX, derived from BY4741 |
| Cell line (*S. cerevisiae*) | LGY0070 | This paper | | HHT1-GFP(S65T)(GGSGGS linker)-HIS3MX, derived from BY4741 |
| Cell line (*S. cerevisiae*) | LGY0071 | This paper | | HHF1-GFP(S65T)(GGSGGS linker)-HIS3MX, derived from BY4741 |
| Antibody | Anti-H2A (Rabbit polyclonal) | ActiveMotif | Cat# 39235, RRID:AB_2687477 | WB (1:1000) |
| Antibody | Anti-H2B (Rabbit polyclonal) | Abcam | Cat# ab1790, RRID:AB_302612 | WB (1:1000) |
| Antibody | Anti-H3 (Rabbit polyclonal) | Abcam | Cat# ab1791, RRID:AB_302613 | WB (1:1000) |
| Antibody | Anti-H4 (Rabbit polyclonal) | Abcam | Cat# ab10158, RRID:AB_296888 | WB (1:1000) |
| Antibody | Anti-GFP (Mouse monoclonal) | Santa Cruz Biotechnology | Cat# sc-9996, RRID:AB_627695 | WB (1:1000) |
| Antibody | Anti-rabbit IgG (Goat polyclonal) | Cell Signaling Technology | Cat# 7074, RRID:AB_2099233 | WB (1:5000) |
| Antibody | Anti-mouse IgG (Goat polyclonal) | Cell Signaling Technology | Cat# 7076, RRID:AB_330924 | WB (1:5000) |
| Recombinant DNA reagent | pFA6a-GFP(S65T)-His3MX6 (plasmid) | Addgene | RRID: Addgene_41598 | GFP tag with Histidine auxotrophy selection marker, contained in DH5-Alpha *Escherichia coli* |
| Recombinant DNA reagent | pFA6a-link-yoTagRFP-T-Kan (plasmid) | Addgene | RRID: Addgene_44906 | G418 resistance selection marker for gene deletion (RFP tag was not used), contained in DH5-Alpha *E. coli* |
| Sequence-based reagent | HTA1-GFP Tag F | This paper | PCR primers for fragment synthesis | AAAGAAGTCTGCCAAGGCT ACCAAGGCTTCTCAAGAATT AAGTAAAGGAGAAGAACTTTT |
| Sequence-based reagent | HTA1-GFP Tag R | This paper | PCR primers for fragment synthesis | TTTAGTTCCTTCCGCCTTCTTTAAAATA CCAGAACCGATCGAATTCGAGCTCGTTTAAAC |
| Sequence-based reagent | HTB1-GFP Tag F | This paper | PCR primers for fragment synthesis | TACTAGAGCTGTTACCAAGTACTCTTCC TCTACTCAAGCAAGTAAAGGAGAAGAACTTTT |

*Appendix 1 Continued on next page*

*Appendix 1 Continued*

| Reagent type (species) or resource | Designation | Source or reference | Identifiers | Additional information |
|---|---|---|---|---|
| Sequence-based reagent | HTB1-GFP Tag R | This paper | PCR primers for fragment synthesis | TAAATAATAATATTAATTATAACCAAAGGA AGTGATTTCAGAATTCGAGCTCGTTTAAAC |
| Sequence-based reagent | HAB2 Del F | This paper | PCR primers for fragment synthesis | AAGAATGTTTGATTTGCTTTGTTTCTTTT CAACTCAGTTCCAGATCCGCTAGGGATAACA |
| Sequence-based reagent | HAB2 Del R | This paper | PCR primers for fragment synthesis | AAAAGAAAACATGACTAAATCACAATA CCTAGTGAGTGACTCGATGAATTCGAGCTCG |
| Sequence-based reagent | HHT1-RIPGLIN-GFP Tag F | This paper | PCR primers for fragment synthesis | GGATATCAAGTTGGCTAGAAGATTAAGA GGTGAAAGATCACGGATCCCCGGGTTAATTAA |
| Sequence-based reagent | HHT1-GFP Tag F | This paper | PCR primers for fragment synthesis | GGATATCAAGTTGGCTAGAAGATTAAGAG GTGAAAGATCAAGTAAAGGAGAAGAACTTTT |
| Sequence-based reagent | HHT1-GFP Tag R | This paper | PCR primers for fragment synthesis | TTTTGTTCGTTTTTTACTAAAACTGATGAC AATCAACAAAGAATTCGAGCTCGTTTAAAC |
| Sequence-based reagent | HHT1-GGSGGS-GFP Tag F | This paper | PCR primers for fragment synthesis | TCCAAAAGAAGGATATCAAGTTGGCTAGA AGATTAAGAGGTGAAAGATCAGGTGGATC TGGTGGATCTAGTAAAGGAGAAGAACTTTT |
| Sequence-based reagent | HHT1-GFP Tag R (Long) | This paper | PCR primers for fragment synthesis | TTTATTGTGTTTTTGTTCGTTTTTTACTAAA ACTGATGACAATCAACAAAGAATTCGAGCTCGTTTA AAC |
| Sequence-based reagent | HHF1-GGSGGS-GFP Tag F | This paper | PCR primers for fragment synthesis | TTGTTTATGCTTTGAAGAGACAAGGTA GAACCTTATACGGTTTCGGTGGTGGT GGATCTGGTGGATCTAGTAAAGGAGAAGAACTTTT |
| Sequence-based reagent | HHF1-GFP Tag R | This paper | PCR primers for fragment synthesis | CGAATCCCAAATATTTGCTTGTTGTT ACCGTTTTCTTAGAATTAGCTAAAGA ATTCGAGCTCGTTTAAAC |
| Sequence-based reagent | FA1 | This paper | PCR primers for confirmation | CGGTGGTAAAGGTGGTAAAG |
| Sequence-based reagent | RA1 | This paper | PCR primers for confirmation | TCGTTTCTGATAAACCAGGT |
| Sequence-based reagent | RG | This paper | PCR primers for confirmation | CCGTTTCATATGATCTGGGT |
| Sequence-based reagent | FH | This paper | PCR primers for confirmation | GACCATTTGCTGTAATCGAC |
| Sequence-based reagent | RK | This paper | PCR primers for confirmation | CCTTATTTTTGACGAGGGGA |
| Sequence-based reagent | RB2 | This paper | PCR primers for confirmation | ATTAACCGGGATTCACTGAC |
| Sequence-based reagent | RA2 | This paper | PCR primers for confirmation | CAGTTCTTGAGAAGCTTTGG |
| Sequence-based reagent | RA2.2 | This paper | PCR primers for confirmation | CTGGACGAAGACGAAGTAAT |
| Sequence-based reagent | FB1 | This paper | PCR primers for confirmation | ATGTCTGCTAAAGCCGAAAA |
| Sequence-based reagent | RB1 | This paper | PCR primers for confirmation | AGTCAGCGACATCTGTCTTT |
| Sequence-based reagent | FT1 | This paper | PCR primers for confirmation | AAGCAAACAGCAAGAAAGTC |
| Sequence-based reagent | RT1 | This paper | PCR primers for confirmation | CTTCTGACAGCAAGGGTATT |
| Sequence-based reagent | FF1 | This paper | PCR primers for confirmation | ATGTCCGGTAGAGGTAAAGG |
| Sequence-based reagent | RF1 | This paper | PCR primers for confirmation | ACACACGAAAATCCTGTGAT |

*Appendix 1 Continued on next page*

*Appendix 1 Continued*

| Reagent type (species) or resource | Designation | Source or reference | Identifiers | Additional information |
|---|---|---|---|---|
| Commercial assay, kit | QIAprep Spin Miniprep Kit (250) | QIAGEN | Cat# 27106 | |
| Commercial assay, kit | QIAquick PCR Purification Kit (50) | QIAGEN | Cat# 28104 | |
| Commercial assay, kit | DNeasy Blood & Tissue Kit (50) | QIAGEN | Cat# 69504 | |
| Commercial assay, kit | Yeast Nuclei Isolation Kit | Abcam | ab206997 | |
| Chemical compound, drug | G418 | Thermo Fisher | 10131035 | (50 mg/mL) |
| Software, algorithm | TFS Tomo4 | Thermo Fisher | | |
| Software, algorithm | Leginon | *Suloway et al., 2009* | RRID:SCR_016731 | |
| Software, algorithm | SerialEM 3.8.6 | *Mastronarde, 2003* | RRID:SCR_017293 | |
| Software, algorithm | PACE-tomo | *Eisenstein et al., 2023* | | |
| Software, algorithm | MotionCor2 | *Zheng et al., 2017* | RRID:SCR_016499 | |
| Software, algorithm | IMOD 4.11 | *Mastronarde, 1997* | RRID:SCR_003297 | |
| Software, algorithm | PEET 1.15 | *Heumann, 2016* | | |
| Software, algorithm | Bsoft 1.8.8 | *Heymann and Belnap, 2007* | RRID:SCR_016503 | |
| Software, algorithm | RELION 3.0.8 | *Kimanius et al., 2016*; *Scheres, 2012* | RRID:SCR_016274 | |
| Software, algorithm | UCSF Chimera 1.13.1 | *Pettersen et al., 2004* | RRID:SCR_002959 | |
| Software, algorithm | Auxilliary cryoEM scripts | *Gan, 2019* | | https://github.com/anaphaze/ot-tools |
| Software, algorithm | Auxilliary cryoEM scripts | *Gaullier, 2021* | | https://github.com/Guillawme/cryoEM-scripts |
| Other | rCutSmart Buffer | New England BioLabs | Cat# B6004 | Buffer for restriction digestion using NEB restriction enzymes |
| Other | SalI Restriction Enzyme | New England BioLabs | Cat# R0138 | Cut plasmids at the restriction site 5'-GTCGAC-3' |
| Other | EcoRV Restriction Enzyme | New England BioLabs | Cat# R0195 | Cut plasmids at the restriction site 5'-GATATC-3' |
| Other | Q5 High-Fidelity 2× Master Mix | New England BioLabs | Cat# M0492 | Contains Q5 DNA Polymerase, deoxynucleotides and $Mg^{2+}$ in buffer, for PCR amplification |
| Other | Salmon sperm DNA | Sigma-Aldrich | D9156 | 10 mg/mL, used for lithium acetate transformation of yeast cells. |
| Other | FloroSafe DNA Stain | Axil Scientific | BIO-5170-1ml | Staining of PCR products in agarose gel after gel electrophoresis |
| Other | 4× Laemmli Sample Buffer | Bio-Rad | #1610747 | Preparation of TCA-precipitated proteins for SDS-PAGE |
| Other | Clarity Western ECL Substrate | Bio-Rad | #1705061 | Visualization of protein bands bound by HRP-conjugated antibodies in immunoblots |

