## [Editor Report · eLife assessment]

This **important** paper exploits new cryo-EM tomography tools to examine the state of chromatin in situ. The experimental work is meticulously performed and **convincing**, with a vast amount of data collected. The main findings are interpreted by the authors to suggest that the majority of yeast nucleosomes lack a stable octameric conformation. Despite the possibly controversial nature of this report, it is our hope that such work will spark thought-provoking debate, and further the development of exciting new tools that can interrogate native chromatin shape and associated function in vivo.

---

## [Referee Report · Reviewer #1 (Public Review)]

This manuscript by Tan et al is using cryo-electron tomography to investigate the structure of yeast nucleosomes both ex vivo (nuclear lysates) and in situ (lamellae and cryosections). The sheer number of experiments and results are astounding and comparable with an entire PhD thesis. However, as is always the case, it is hard to prove that something is not there. In this case, canonical nucleosomes. In their path to find the nucleosomes, the authors also stumble over new insights into nucleosome arrangement that indicates that the positions of the histones is more flexible than previously believed.

Major strengths and weaknesses:

Personally, I am not ready to agree with their conclusion that heterogenous non-canonical nucleosomes predominate in yeast cells, but this reviewer is not an expert in the field of nucleosomes and can't judge how well these results fit into previous results in the field. As a technological expert though, I think the authors have done everything possible to test that hypothesis with today's available methods. One can debate whether it is necessary to have 35 supplementary figures, but after working through them all, I see that the nature of the argument needs all that support, precisely because it is so hard to show what is not there. The massive amount of work that has gone into this manuscript and the state-of-the art nature of the technology should be warmly commended. I also think the authors have done a really great job with including all their results to the benefit of the scientific community. Yet, I am left with some questions and comments:

Could the nucleosomes change into other shapes that were predetermined in situ? Could the authors expand on if there was a structure or two that was more common than the others of the classes they found? Or would this not have been found because of the template matching and later reference particle used?

Could it simply be that the yeast nucleoplasm is differently structured than that of HeLa cells and it was harder to find nucleosomes by template matching in these cells? The authors argue against crowding in the discussion, but maybe it is just a nucleoplasm texture that side-tracks the programs?

The title of the paper is not well reflected in the main figures. The title of Figure 2 says "Canonical nucleosomes are rare in wild-type cells", but that is not shown/quantified in that figure. Rare is comparison to what? I suggest adding a comparative view from the HeLa cells, like the text does in lines 195-199. A measure of nucleosomes detected per volume nucleoplasm would also facilitate a comparison.

If the cell contains mostly non-canonical nucleosomes, are they really non-canonical? Maybe a change of language is required once this is somewhat sure (say, after line 303).

The authors could explain more why they sometimes use conventional the 2D followed by 3D classification approach and sometimes "direct 3-D classification". Why, for example, do they do 2D followed by 3D in Figure S5A? This Figure could be considered a regular figure since it shows the main message of the paper.

Figure 1: Why is there a gap in the middle of the nucleosome in panel B? The authors write that this is a higher resolution structure (18Å), but in the even higher resolution crystallography structure (3Å resolution), there is no gap in the middle.

---

## [Referee Report · Reviewer #2 (Public Review)]

Nucleosome structures inside cells remain unclear. Tan et al. tackled this problem using cryo-ET and 3-D classification analysis of yeast cells. The authors found that the fraction of canonical nucleosomes in the cell could be less than 10% of total nucleosomes. The finding is consistent with the unstable property of yeast nucleosomes and the high proportion of the actively transcribed yeast genome. The authors made an important point in understanding chromatin structure in situ. Overall, the paper is well-written and informative to the chromatin/chromosome field.

---

## [Referee Report · Reviewer #3 (Public Review)]

Several labs in the 1970s published fundamental work revealing that almost all eukaryotes organize their DNA into repeating units called nucleosomes, which form the chromatin fiber. Decades of elegant biochemical and structural work indicated a primarily octameric organization of the nucleosome with 2 copies of each histone H2A, H2B, H3 and H4, wrapping 147bp of DNA in a left handed toroid, to which linker histone would bind.

This was true for most species studied (except, yeast lack linker histone) and was recapitulated in stunning detail by in vitro reconstitutions by salt dialysis or chaperone-mediated assembly of nucleosomes. Thus, these landmark studies set the stage for an exploding number of papers on the topic of chromatin in the past 45 years.

An emerging counterpoint to the prevailing idea of static particles is that nucleosomes are much more dynamic and can undergo spontaneous transformation. Such dynamics could arise from intrinsic instability due to DNA structural deformation, specific histone variants or their mutations, post-translational histone modifications which weaken the main contacts, protein partners, and predominantly, from active processes like ATP-dependent chromatin remodeling, transcription, repair and replication.

This paper is important because it tests this idea whole-scale, applying novel cryo-EM tomography tools to examine the state of chromatin in yeast lysates or cryo-sections. The experimental work is meticulously performed, with vast amount of data collected. The main findings are interpreted by the authors to suggest that majority of yeast nucleosomes lack a stable octameric conformation. The findings are not surprising in that alternative conformations of nucleosomes might exist in vivo, but rather in the sheer scale of such particles reported, relative to the traditional form expected from decades of biochemical, biophysical and structural data. Thus, it is likely that this work will be perceived as controversial. Nonetheless, we believe these kinds of tools represent an important advance for in situ analysis of chromatin. We also think the field should have the opportunity to carefully evaluate the data and assess whether the claims are supported, or consider what additional experiments could be done to further test the conceptual claims made. It is our hope that such work will spark thought-provoking debate in a collegial fashion, and lead to the development of exciting new tools which can interrogate native chromatin shape in vivo. Most importantly, it will be critical to assess biological implications associated with more dynamic - or static forms- of nucleosomes, the associated chromatin fiber, and its three-dimensional organization, for nuclear or mitotic function.

---

## [Author Response]

**eLife assessment**
This important paper exploits new cryo-EM tomography tools to examine the state of chromatin in situ. The experimental work is meticulously performed and convincing, with a vast amount of data collected. The main findings are interpreted by the authors to suggest that the majority of yeast nucleosomes lack a stable octameric conformation. Despite the possibly controversial nature of this report, it is our hope that such work will spark thought-provoking debate, and further the development of exciting new tools that can interrogate native chromatin shape and associated function in vivo.

We thank the Editors and Reviewers for their thoughtful and helpful comments. We also appreciate the extraordinary amount of effort needed to assess both the lengthy manuscript and the previous reviews. Below, we provide our provisional responses in bold blue font. The majority of the comments are straightforward to address. We have taken a more conservative approach with the subset of comments that would require us to speculate because we either lack key information or we lack technical expertise. Instead of adding the speculative replies to the main text, we think it will be better to leave them in the rebuttal for posterity. Readers will therefore have access to our speculation and know that we did not feel confident enough to include these thoughts in the Version of Record.

**Reviewer #1 (Public Review):**
This manuscript by Tan et al is using cryo-electron tomography to investigate the structure of yeast nucleosomes both ex vivo (nuclear lysates) and in situ (lamellae and cryosections). The sheer number of experiments and results are astounding and comparable with an entire PhD thesis. However, as is always the case, it is hard to prove that something is not there. In this case, canonical nucleosomes. In their path to find the nucleosomes, the authors also stumble over new insights into nucleosome arrangement that indicates that the positions of the histones is more flexible than previously believed.

We want to point out that canonical nucleosomes are there in wild-type cells in situ, albeit rarer than what’s expected based on our HeLa cell analysis. The negative result (absence of any canonical nucleosome classes in situ) was found in the histone-GFP mutants.

Major strengths and weaknesses:Personally, I am not ready to agree with their conclusion that heterogenous non-canonical nucleosomes predominate in yeast cells, but this reviewer is not an expert in the field of nucleosomes and can't judge how well these results fit into previous results in the field. As a technological expert though, I think the authors have done everything possible to test that hypothesis with today's available methods. One can debate whether it is necessary to have 35 supplementary figures, but after working through them all, I see that the nature of the argument needs all that support, precisely because it is so hard to show what is not there. The massive amount of work that has gone into this manuscript and the state-of-the art nature of the technology should be warmly commended. I also think the authors have done a really great job with including all their results to the benefit of the scientific community. Yet, I am left with some questions and comments:Could the nucleosomes change into other shapes that were predetermined in situ? Could the authors expand on if there was a structure or two that was more common than the others of the classes they found? Or would this not have been found because of the template matching and later reference particle used?

Our best guess (speculation) is that one of the class averages that is smaller than the canonical nucleosome contains one or more non-canonical nucleosome classes. We do not feel confident enough to single out any of these classes precisely because we do not yet know if they arise from one non-canonical nucleosome structure or from multiple – and therefore mis-classified – non-canonical nucleosome structures (potentially with other non-nucleosome complexes mixed in). We feel it is better to leave this discussion out of the manuscript, or risk sending the community on wild goose chases.

Our template-matching workflow uses a low-enough cross-correlation threshold that any nucleosome-sized particle (plus minus a few nanometers) would be picked, which is why the number of hits is so large. So unless the noncanonical nucleosomes quadrupled in size or lost most of their histones, they should be grouped with one or more of the other 99 class averages (WT cells) or any of the 100 class averages (cells with GFP-tagged histones). As to whether the later reference particle could have prevented us from detecting one of the non-canonical nucleosome structures, we are unable to tell because we’d really have to know what an in situ non-canonical nucleosome looks like first.

Could it simply be that the yeast nucleoplasm is differently structured than that of HeLa cells and it was harder to find nucleosomes by template matching in these cells? The authors argue against crowding in the discussion, but maybe it is just a nucleoplasm texture that side-tracks the programs?

Presumably, the nucleoplasmic “side-tracking” texture would come from some molecules in the yeast nucleus. These molecules would be too small to visualize as discrete particles in the tomographic slices, but they would contribute textures that can be “seen” by the programs – in particular RELION, which does the discrimination between structural states. We do not know the inner-workings of RELION well enough to say what kinds of density textures would side-track its classification routines.

The title of the paper is not well reflected in the main figures. The title of Figure 2 says "Canonical nucleosomes are rare in wild-type cells", but that is not shown/quantified in that figure. Rare is comparison to what? I suggest adding a comparative view from the HeLa cells, like the text does in lines 195-199. A measure of nucleosomes detected per volume nucleoplasm would also facilitate a comparison.

Figure 2’s title is indeed unclear and does not align with the paper’s title and key conclusion. The rarity here is relative to the expected number of nucleosomes (canonical plus non-canonical). We have changed the title to “Canonical nucleosomes are a minority of the expected total in wild-type cells”. We would prefer to leave the reference to HeLa cells to the main text instead of as a figure panel because the comparison is not straightforward for a graphical presentation. Instead, we will report the total number of nucleosomes estimated for this particular tomogram (~7,600) versus the number of canonical nucleosomes classified (297; 594 if we assume we missed half of them).

If the cell contains mostly non-canonical nucleosomes, are they really non-canonical? Maybe a change of language is required once this is somewhat sure (say, after line 303).

This is an interesting semantic and philosophical point. From the yeast cell’s “perspective”, the canonical nucleosome structure would be the form that is in the majority. That being said, we do not know if there is one structure that is the majority. From the chromatin field’s point of view, the canonical nucleosome is the form that is most commonly seen in all the historical – and most contemporary – literature, namely something that resembles the crystal structure of Luger et al, 1997. Given these two lines of thinking, we will add the following clarification after line 303:

“At present, we do not know what the non-canonical nucleosome structures are, meaning that we cannot even determine if one non-canonical structure is the majority. Until we know what the family of non-canonical nucleosome structures are, we will use the term non-canonical to describe the nucleosomes that do not have the canonical (crystal) structure”.

The authors could explain more why they sometimes use conventional the 2D followed by 3D classification approach and sometimes "direct 3-D classification". Why, for example, do they do 2D followed by 3D in Figure S5A? This Figure could be considered a regular figure since it shows the main message of the paper.

Because the classification of subtomograms in situ is still a work in progress, we felt it would be better to show one instance of 2-D classification for lysates and one for lamellae. While it is true that we could have presented direct 3-D classification for the entire paper, we anticipate that readers will be interested to see what the in situ 2-D class averages look like.

The main message is that there are canonical nucleosomes in situ (at least in wild-type cells), but they are a minority. Therefore, the conventional classification for Figure S5A should not be a main figure because it does not show any canonical nucleosome class averages in situ.

Figure 1: Why is there a gap in the middle of the nucleosome in panel B? The authors write that this is a higher resolution structure (18Å), but in the even higher resolution crystallography structure (3Å resolution), there is no gap in the middle.

There is a lower concentration of amino acids at the middle in the disc view; unfortunately, the space-filling model in Figure 1A hides this feature. The gap exists in experimental cryo-EM density maps. See below for an example. The size of the gap depends on the contour level and probably the contrast mechanism, as the gap is less visible in the VPP subtomogram averages. To clarify this confusing phenomenon, we will add the following lines to the figure legend:

“The gap in the disc view of the nuclear-lysate-based average is due to the lower concentration of amino acids there, which is not visible in panel A due to space-filling rendering. This gap’s size may depend on the contrast mechanism because it is not visible in the VPP averages.”

**Author response image 1. sa4fig1:** 

**Reviewer #2 (Public Review):**
Nucleosome structures inside cells remain unclear. Tan et al. tackled this problem using cryo-ET and 3-D classification analysis of yeast cells. The authors found that the fraction of canonical nucleosomes in the cell could be less than 10% of total nucleosomes. The finding is consistent with the unstable property of yeast nucleosomes and the high proportion of the actively transcribed yeast genome. The authors made an important point in understanding chromatin structure in situ. Overall, the paper is well-written and informative to the chromatin/chromosome field.

We thank Reviewer 2 for their positive assessment.

**Reviewer #3 (Public Review):**
Several labs in the 1970s published fundamental work revealing that almost all eukaryotes organize their DNA into repeating units called nucleosomes, which form the chromatin fiber. Decades of elegant biochemical and structural work indicated a primarily octameric organization of the nucleosome with 2 copies of each histone H2A, H2B, H3 and H4, wrapping 147bp of DNA in a left handed toroid, to which linker histone would bind.This was true for most species studied (except, yeast lack linker histone) and was recapitulated in stunning detail by in vitro reconstitutions by salt dialysis or chaperone-mediated assembly of nucleosomes. Thus, these landmark studies set the stage for an exploding number of papers on the topic of chromatin in the past 45 years.An emerging counterpoint to the prevailing idea of static particles is that nucleosomes are much more dynamic and can undergo spontaneous transformation. Such dynamics could arise from intrinsic instability due to DNA structural deformation, specific histone variants or their mutations, post-translational histone modifications which weaken the main contacts, protein partners, and predominantly, from active processes like ATP-dependent chromatin remodeling, transcription, repair and replication.This paper is important because it tests this idea whole-scale, applying novel cryo-EM tomography tools to examine the state of chromatin in yeast lysates or cryo-sections. The experimental work is meticulously performed, with vast amount of data collected. The main findings are interpreted by the authors to suggest that majority of yeast nucleosomes lack a stable octameric conformation. The findings are not surprising in that alternative conformations of nucleosomes might exist in vivo, but rather in the sheer scale of such particles reported, relative to the traditional form expected from decades of biochemical, biophysical and structural data. Thus, it is likely that this work will be perceived as controversial. Nonetheless, we believe these kinds of tools represent an important advance for in situ analysis of chromatin. We also think the field should have the opportunity to carefully evaluate the data and assess whether the claims are supported, or consider what additional experiments could be done to further test the conceptual claims made. It is our hope that such work will spark thought-provoking debate in a collegial fashion, and lead to the development of exciting new tools which can interrogate native chromatin shape in vivo. Most importantly, it will be critical to assess biological implications associated with more dynamic - or static forms- of nucleosomes, the associated chromatin fiber, and its three-dimensional organization, for nuclear or mitotic function.

Thank you for putting our work in the context of the field’s trajectory. We hope our EMPIAR entry, which includes all the raw data used in this paper, will be useful for the community. As more labs (hopefully) upload their raw data and as image-processing continues to advance, the field will be able to revisit the question of non-canonical nucleosomes in budding yeast and other organisms.